# UniSVD: Unilateral Weight Decomposition for Attention-based Vision Models

## Abstract

Transformers have achieved remarkable success across diverse domains, but their ever-growing scale results in prohibitive computational and memory costs. Low-rank matrix decomposition with Singular Value Decomposition (SVD) has emerged as an effective compression technique. Recent studies, such as ASVD, SVD-LLM, and FLAR-SVD, have improved decomposition quality by incorporating activation-aware method. However, these methods do not consider the unique mechanism of MHA, where query-key ($Q$-$K$) and value–output ($V$-$O$) computations are linear and allow pre-computation. To effectively leverage this mechanism, we propose Unilateral Singular Value Decomposition (UniSVD), a novel framework that applies decomposition to only one side of the $Q$-$K$ or $V$-$O$ weight pairs in a head-wise manner. Since $Q$-$K$-$V$-$O$ weights exhibit varying sensitivities to low-rank approximation across heads and layers, UniSVD adaptively selects which side to be decomposed according to the rank sensitivity, thereby preserving the important information of weights. Extensive experiments demonstrate that UniSVD seamlessly integrates with existing decomposition methods and consistently achieves superior trade-offs between parameter reduction, FLOPs, and model performance.

## 1 Introduction

Transformers (Vaswani et al., 2017; Dosovitskiy et al., 2020; Touvron et al., 2021; Radford et al., 2021; Touvron et al., 2022; Liu et al., 2021a; Wang et al., 2021) have achieved remarkable success across a wide range of applications. Their ability to model long-range dependencies through the attention mechanism has established them as a foundational architecture in modern deep learning. However, as the scale of Transformer models continues to grow, they pose significant challenges in terms of computation, latency, and efficiency. To address these challenges, model compression techniques have been extensively explored, such as quantization, model distillation, pruning, and matrix decomposition. This study focuses on matrix decomposition methods that do not rely on backward propagation as required in recovery fine-tuning, while still enabling efficient inference.

For matrix decomposition (Wang et al., 2025a; Yang et al., 2024a; Saha et al., 2023; Qin et al., 2025; Pletenev et al., 2023), low-rank factorization with Singular Value Decomposition (SVD) offers an promising approach for compressing transformer-based models (Wu et al., 2023; Yu & Wu, 2023; Ashkboos et al., 2024; Chang et al., 2025). The conventional SVD method factorizes each weight matrix into two low-rank matrices while preserving essential information of original weight. Recent studies(Tian et al., 2025; Chen et al., 2021) have investigated activation-guided weight importance to enhance decomposition quality. For example, ASVD (Yuan et al., 2023) introduces a diagonal scaling matrix to mitigate the distribution shift in activations before and after truncation. SVD-LLM (Wang et al., 2024; 2025b) apply a truncation-aware whitening strategy that attains the theoretical minimum loss. To enhance stability, FLAR-SVD (Thoma et al., 2025) integrates a Ledoit-Wolf shrinkage estimator into Cholesky SVD framework and applies it to vision models.

However, existing per-weight decomposition exhibit an inherent inefficiency in attention where each Query ($Q$), Key ($K$), Value ($V$), and Output ($O$) weights ($W^Q, W^K, W^V, W^O \in \mathbb{R}^{d \times d}$) is factorized into two low-rank matrices ($U \in \mathbb{R}^{d \times r}$, $S \in \mathbb{R}^{r \times d}$) and used as two sublayers. As a result, substantial efficiency gains can only be realized when the reduced rank is smaller than $2/d$. To address this issue, ModeGPT (Lin et al., 2024) introduces a column-selection-based CR decomposition (Drineas et al., 2006) for the $W^Q$-$W^K$ pair, while applying SVD to both sides of the

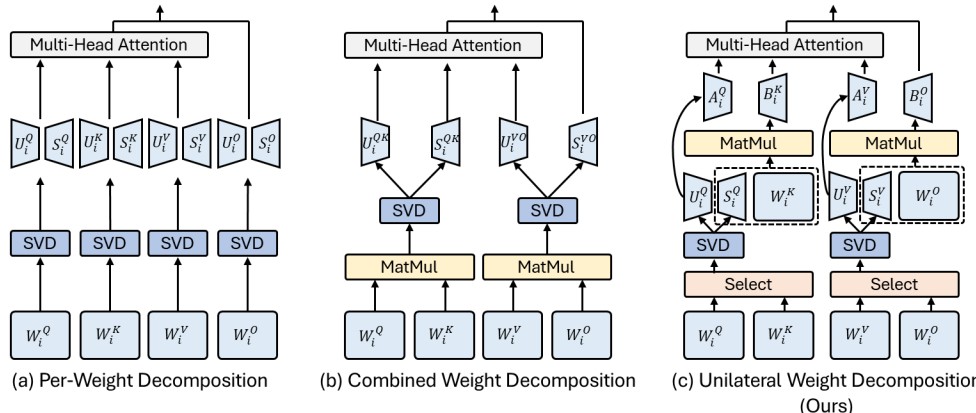

Figure 1: Comparison of the different weight decomposition methods in multi-head attention: (a) Per-weight decomposition, (b) Combined weight decomposition, and (c) Unilateral weight decomposition (UniSVD). The illustration of UniSVD is shown for the case of $W^Q$ and $W^V$ be selected.

$W^V$-$W^O$ weights and recombining the decomposed matrices through matrix multiplication. This method alleviates part of the inefficiency inherent in conventional factorization. Nevertheless, it still decomposes each weight matrix, which insufficient to prevent performance degradation.

When the weight decomposition is applied to Transformer-based vision models, these methods do not consider the fundamental linear characteristic of Multi-Head Attention (MHA). In MHA, the input is projected into $Q$, $K$ and $V$ representations, followed by the scaled dot-product operation and an $O$ projection, all performed within a multi-head structure. A notable characteristic of this process is that the $Q$-$K$ and $V$-$O$ computations involve no intermediate non-linear functions. COMCAT (Xiao et al., 2023) successfully incorporates this characteristic to addresses the inefficiency while avoiding per-weight matrix decomposition by introducing a combined weight decomposition strategy, which firstly multiplicates each $W^Q$-$W^K$ and $W^V$-$W^O$ and decomposes the combined weight matrix.

In this paper, we propose Unilateral Singular Value Decomposition (UniSVD), a novel and effective method that applies decomposition to only one side of the $Q$-$K$ and $V$-$O$ weights in a head-wise manner. Our approach addresses the inefficiency of per-weight decomposition methods while demonstrating superior performance preservation compared to combined decomposition methods, particularly under low-rank approximation. When decomposing unilaterally, we focus on that each weight within the $Q$-$K$ and $V$-$O$ pairs exhibits different sensitivities to low-rank approximation across heads and layers. Therefore, as illustrated in Figure 1 (c), we dynamically select which weight matrix to decompose based on the approximation error. After decomposing one side of the weight matrices, we multiply it with the other side, thereby reducing the overall weight while preserving the information of the non-decomposed side, and still achieving the effect of decomposing both sides.

This design choice is motivated by the analysis presented in Figure 2. We analyze the singular value distributions of the individual weights (i.e., $W^Q$, $W^K$, $W^V$ and $W^O$) and the combined weights (i.e., $W^{QK}$ and $W^{VO}$), where singular values are proportional to truncation error. The results reveal that the combined weights exhibit noticeably larger singular values in the low-rank regime compared to the individual weight matrices. This observation indicates that applying decomposition optimized for individual weights can be more effective than the combined weight decomposition, which is particularly vulnerable to performance degradation under low-rank approximation.

Our UniSVD provides a robust and generalizable framework for weight compression in attention structures. Experimental results demonstrate that UniSVD can be seamlessly integrated with various decomposition techniques, consistently delivering superior trade-offs among FLOPs, parameter reduction, and model performance. Our contributions are summarized as follows:

- We propose Unilateral Singular Value Decomposition (UniSVD), a novel method that decomposes only one side of the $Q$-$K$ and $V$-$O$ weights, effectively mitigating performance degradation commonly observed in low-rank approximations.

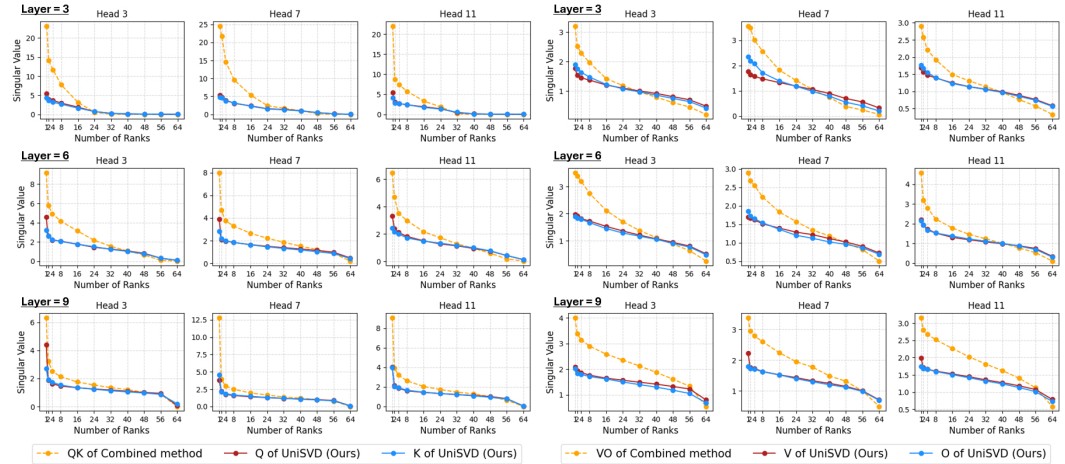

Figure 2: Analysis of the singular value distributions at each rank for the combined weight matrices and the individual weight matrices to compare the sensitivity to low-rank approximation.

- UniSVD leverages a dynamic selection strategy, where the decomposed side is determined head-wise based on low-rank approximation error.
- We demonstrate that UniSVD can be seamlessly integrated into the attention decomposition component of various decomposition methods, consistently achieving significant performance improvements and latency reductions.

## 2 RELATED WORKS

### 2.1 MODEL COMPRESSION FOR TRANSFORMERS

Transformers have demonstrated remarkable performance across natural language processing, vision, and multimodal tasks, but these advances incur substantial resource demands. Various model compression techniques have been extensively explored to reduce the computations and accelerate the models. Knowledge distillation (Pan et al., 2020; Fang et al., 2021; Zhao et al., 2023) transfers behaviors from the complex teacher model to the compact student model via the softened logits, intermediate representation matching, and attention alignments. Pruning methods (LeCun et al., 1989; Zhu et al., 2021; Frantar & Alistarh, 2023; Yang et al., 2023; Gao et al., 2024b) set the less important weights to zero or remove the redundant components, such as weights, neurons or tokens. Quantization methods (Courbariaux et al., 2015; Liu et al., 2021b; Shang et al., 2023; Lv et al., 2024) compress the numerical model parameters from high-precision floating-point numbers to lower-precision integers. Recently, low-rank matrix decomposition (Kim et al., 2015; Wen et al., 2017; Schotthöfer et al., 2022; Li et al., 2023; Chang et al., 2024a) has been actively explored for model compression, which approximates high-dimensional weight matrices with lower-rank matrices. Our study focuses on training-free matrix decomposition techniques for efficient model compression.

### 2.2 WEIGHT DECOMPOSITION FOR NATURAL LANGUAGE PROCESSING

Weight Decomposition (Makni et al., 2025; Gu et al., 2025; Smith et al., 2025; Chein et al., 2024; Gao et al., 2024a; Yang et al., 2024b; Ji et al., 2024b; Hua et al., 2025; Huang et al., 2025) has recently emerged as a promising approach for compressing and accelerating Transformers(Ji et al., 2024a; Lee et al., 2024; Anjum et al., 2024; Li et al., 2022; Saha et al., 2024; Sharma et al., 2023). FWSVD (Hsu et al., 2022) introduced Fisher-weighted decomposition that leverages curvature information to preserve critical directions during truncation. ASVD (Yuan et al., 2023) proposed a diagonal scaling matrix, which scales the weight matrix based on the distribution patterns of input activation channels. More recent methods, SVD-LLM (Wang et al., 2024) and SVD-LLM V2 (Wang et al., 2025b), introduced a truncation-aware data whitening scheme to address the misalignment between

the singular value truncation and the compression loss. ModeGPT (Lin et al., 2024) proposed a modular decomposition that jointly decomposes multiple matrices within each module by using SVD, Nyström (Gittens & Mahoney, 2016) and CR decomposition (Drineas et al., 2006) .

## 2.3 Weight Decomposition for Vision Models

As demonstrated in NLP, the weight decomposition has also been recently studied for vision models(Veeramacheneni et al.; Chang et al., 2024b; Jie & Deng, 2023). PELA (Guo et al., 2024) exploited low-rank approximation to entirely replace the pre-trained weights with the reduced low-rank matrices for downstream fine-tuning tasks. Azizi et al. (2024) proposed an activation-aware mixed-rank compression strategy that adaptively allocates low-rank factors based on activation sensitivity in vision transformers. Several studies (Thoma et al., 2025; Xiao et al., 2023) explored the weight decomposition without fine-tuning for vision models. FLAR-SVD (Thoma et al., 2025) used the Ledoit-Wolf shrinkage estimator for a stabilized Cholesky SVD. By exploiting the characteristic of the attention in vision models, COMCAT (Xiao et al., 2023) proposed combined weight decomposition method which factorized the combined weight matrices for head-level low-rankness. Our study also exploits the characteristics of attention in vision models but introduces a novel approach that dynamically selects and decomposes only one side of the weight pairs.

## 3 Method

Since our approach leverages linear characteristics of the Query($Q$)-Key($K$) and Value($V$)-Output($O$) operations within Multi-Head Attention (MHA), we first provide an overview of the multi-head attention mechanism. Then, we discuss existing decomposition methods that exploit this property, followed by a detailed introduction of the proposed Unilateral Singular Value Decomposition (UniSVD).

## 3.1 Reformulation of Multi-Head Attention

Our weight decomposition method leverages the linear transformation property inherent in the $Q$-$K$ and $V$-$O$ operations of multi-head attention (MHA), a characteristic that clearly differentiates it from convolutional and feedforward layers. the MHA mechanism, given an input $X \in \mathbb{R}^{N \times C}$, the per-head projections $Q_i$, $K_i$, and $V_i$ are generated through their respective weight matrices $W_i^Q$, $W_i^K$, and $W_i^V \in \mathbb{R}^{C \times d_h}$. Following this step, the head-wise scaled dot-product attention is computed to obtain $head_i$, and finally, the concatenated outputs from all heads are passed through the projection matrix $W^O \in \mathbb{R}^{C \times C}$, yielding the final output representation. This can be formulated as follows.

$$\text{head}_i = \text{Attention}(XW_i^Q, XW_i^K, XW_i^V) = \text{Softmax}\left(\frac{XW_i^Q(XW_i^K)^\top}{\sqrt{d_h}}\right)XW_i^V, \quad (1)$$

$$\text{MHA}(X) = \text{Concat}(\text{head}_1, \ldots, \text{head}_h)W^O, \quad (2)$$

where Concat denotes the concatenation operator along the head dimension, $d_h$ represents the dimension of each head, and $h$ is the total number of heads. To explicitly describe the head-wise operations among $W_i^Q, W_i^K, W_i^V$, and $W^O$, we reformulate MHA such that the output projection $W^O$ is divided across heads, with each head associated with $W_i^O \in \mathbb{R}^{d_h \times C}$, as follows:

$$\text{MHA}(X) = \sum_{i=1}^{h} \text{head}_i W_i^O = \sum_{i=1}^{h} \text{Softmax}\left(\frac{X(W_i^Q W_i^{K\top})X^\top}{\sqrt{d_h}}\right)X(W_i^V W_i^O), \quad (3)$$

Therefore, MHA can be reformulated as a series of matrix multiplications between $W_i^Q$-$W_i^K$ and $W_i^V$-$W_i^O$ for the linear transformations of each head.

## 3.2 Conventional SVD for Attention

Based on the formulation of MHA in Eq. (3), the conventional per-weight decomposition method performs low-rank approximation by applying SVD to each of the projection matrices $W_i^Q, W_i^K, W_i^V$,

and $W_i^O$. The SVD is employed to decompose the weight matrix $W \in \mathbb{R}^{in \times out}$ into a low-rank factorization of rank $r$, expressed as $W \approx U\Sigma S'$, where $U \in \mathbb{R}^{in \times r}$, $S' \in \mathbb{R}^{r \times out}$, and $\Sigma \in \mathbb{R}^{r \times r}$ is a diagonal matrix containing the singular values. Subsequently, $\Sigma$ and $S'$ are multiplied to form $S = \Sigma S'$, allowing the decomposition to be represented using two matrices, $U$ and $S$, which are then used for the low-rank approximation. As shown in Figure 1 (a), performing the naive SVD for low-rank approximation on each weight matrix in MHA can be formulated as follows:

$$\text{MHA}(X) \approx \sum_{i=1}^{h} \text{Softmax}\left(\frac{X(U_i^Q S_i^Q S_i^{K\top} U_i^{K\top})X^\top}{\sqrt{d_h}}\right) X(U_i^V S_i^V U_i^O S_i^O), \qquad (4)$$

where $U_i^Q, U_i^K, U_i^V \in \mathbb{R}^{C \times r}$ and $U_i^O \in \mathbb{R}^{d_h \times r}$ denote the left singular vector matrices obtained from SVD, $S_i^Q, S_i^K, S_i^V \in \mathbb{R}^{r \times d_h}$ and $S_i^O \in \mathbb{R}^{r \times C}$ are the corresponding low-rank factor matrices. As shown in Figure 1 (b), combined decomposition method (Xiao et al., 2023) first performs matrix multiplication between $W_i^Q$-$W_i^K$ and $W_i^V$-$W_i^O$, and then applies SVD to effectively allocate the decomposed weights. The low-rank factorized MHA can be formulated as follows:

$$\text{MHA}(X) \approx \sum_{i=1}^{h} \text{Softmax}\left(\frac{X(U_i^{QK} S_i^{QK\top})X^\top}{\sqrt{r}}\right) X(U_i^{VO} S_i^{VO}), \qquad (5)$$

where $U_i^{QK}$ and $S_i^{QK} \in \mathbb{R}^{C \times r}$ denote the low-rank factors obtained from the head-wise SVD of $W_i^Q W_i^{K\top}$. $U_i^{VO} \in \mathbb{R}^{C \times r}$ and $S_i^{VO} \in \mathbb{R}^{r \times C}$ represent the low-rank factors obtained from the head-wise SVD of $W_i^V W_i^O$. These low-rank factors, $U_i^{QK}, S_i^{QK}, U_i^{VO}$ and $S_i^{VO}$, are used as the compressed projection weights for $Q$, $K$, $V$ and $O$, respectively.

### 3.3 UNILATERAL SINGULAR VALUE DECOMPOSITION

The proposed Unilateral Singular Value Decomposition (UniSVD) selectively decomposes only one side of the weight pairs, $W_i^Q$-$W_i^K$ and $W_i^V$-$W_i^O$, while achieving a more effective approximation than applying low-rank factorization to both sides. Since the rank sensitivity of these pairs varies across heads and layers, we first evaluate the relative sensitivity by computing a Frobenius norm-based loss. This measure allows us to identify the less rank-sensitive weight within each pair, thereby guiding the selection of the side to be decomposed. The Frobenius norm-based loss is defined as:

$$\text{SVD}(W) = \tilde{W}(r) = US, \ L_I(W, r) = ||W - \tilde{W}(r)||_F, \qquad (6)$$

where $\tilde{W}(r)$ denotes the rank $r$ approximation of $W$, obtained via truncated SVD. This formulation provides a principled criterion for selecting the decomposition side. Based on this criterion, we determine which weight to be decomposed within each pair, either $W_i^Q$ or $W_i^K$, and $W_i^V$ or $W_i^O$. After that, we apply SVD only to the selected weight as shown in Figure 1 (c). The resulting low-rank factorization is then combined with its non-decomposed counterpart via a linear operator. This enables the model to preserve information from the intact side while reducing the computational complexity on the decomposed side. This process is formulated as follows:

$$\delta_i^{\text{QK}} = \mathbb{1}\left[\ell(W_i^Q, r_{QK}) \le \ell(W_i^K, r_{QK})\right], \qquad \beta_i^{\text{VO}} = \mathbb{1}\left[\ell(W_i^V, r_{VO}) \le \ell(W_i^O, r_{VO})\right], \qquad (7)$$

$$A_i^Q = \delta_i^{\text{QK}} U_i^Q + (1 - \delta_i^{\text{QK}})W_i^Q S_i^{K\top}, \ B_i^K = \delta_i^{\text{QK}} S_i^Q W_i^{K\top} + (1 - \delta_i^{\text{QK}})U_i^{K\top}, \qquad (8)$$

$$A_i^V = \beta_i^{\text{VO}} U_i^V + (1 - \beta_i^{\text{VO}})W_i^V S_i^{O\top}, \ B_i^O = \beta_i^{\text{VO}} S_i^V W_i^{O\top} + (1 - \beta_i^{\text{VO}})U_i^{O\top}, \qquad (9)$$

where $A_i^Q \in \mathbb{R}^{C \times r_{QK}}$, $B_i^K \in \mathbb{R}^{r_{QK} \times C}$, $A_i^V \in \mathbb{R}^{C \times r_{VO}}$, and $B_i^O \in \mathbb{R}^{r_{VO} \times C}$ denote the selectively compressed weights. These matrices are used as the effective projection weights for $Q$, $K$, $V$, and $O$, respectively. $r_{QK}$ and $r_{VO}$ represent the ranks for low-rank approximation. In practice, we observed that the rank sensitivity of the $W_i^Q$-$W_i^K$ pair differs from that of the $W_i^V$-$W_i^O$ pair. Therefore, we assign different target ranks for each pair and further explore a simple hierarchical strategy that adapts the rank settings across layers. Given a predefined pair $(r_{min}, r_{max})$, the ranks are assigned layer-wise according to an arithmetic progression between these two values. Accordingly, the low-rank factorized MHA with UniSVD is formulated as follows:

$$\text{MHA}(X) \approx \sum_{i=1}^{h} \text{Softmax}\left(\frac{X(A_i^Q B_i^K)X^\top}{\sqrt{r_{QK}}}\right) X(A_i^V B_i^O). \qquad (10)$$

| Method | DeiT-Small | | | | DeiT-Base | | | |
|---|---|---|---|---|---|---|---|---|
| | Params (M) | GFLOPs | Top-1 ↑ | Latency ↓ | Params (M) | GFLOPs | Top-1 ↑ | Latency ↓ |
| Base | 22.1 | 8.5 | 79.9 | 11.3 | 86.6 | 33.7 | 81.8 | 30.7 |
| FWSVD (Hsu et al., 2022) | 15.6 | 6.0 | 62.5 | 11.7 | 44.0 | 16.9 | 73.5 | 25.6 |
| FWSVD + UniSVD (**Ours**) | 15.7 | 6.0 | **68.8** | **11.0** | 44.1 | 17.0 | **73.7** | **23.6** |
| ASVD (Yuan et al., 2023) | 15.6 | 6.0 | 58.6 | 11.7 | 44.1 | 17.0 | 71.8 | 26.1 |
| ASVD + UniSVD (**Ours**) | 15.7 | 6.0 | **67.1** | **10.8** | 44.1 | 17.0 | **72.4** | **23.9** |
| SVD-LLM (Wang et al., 2024) | 15.6 | 6.0 | 66.9 | 12.1 | 44.1 | 17.0 | 70.3 | 26.7 |
| SVD-LLM + UniSVD (**Ours**) | 15.4 | 5.9 | **69.7** | **11.3** | 44.1 | 17.0 | **72.2** | **25.1** |
| FLAR-SVD (Thoma et al., 2025) | 15.7 | 6.0 | 66.5 | 10.8 | 49.2 | 19.0 | 78.9 | 26.5 |
| FLAR-SVD + UniSVD (**Ours**) | 16.0 | 6.1 | **69.3** | **10.2** | 49.4 | 19.1 | **79.3** | **23.5** |

Table 1: Accuracy and latency comparison of UniSVD against various weight decomposition baselines on ImageNet-1K without fine-tuning.

This formulation ensures that the model retains information on the intact side while achieving the effect of decomposing both sides. Compared to the naive SVD and the combined SVD approaches, our UniSVD leads to a better performance-efficiency trade-off with respect to the number of parameters and FLOPs due to a more effective decomposition strategy.

## 4 EXPERIMENTS

### 4.1 EXPERIMENTAL SETTINGS

We evaluate our approach on ImageNet-1K (Deng et al., 2009) dataset and adopt DeiT (Touvron et al., 2021) series as baseline models, which are widely adopted by previous SVD-based methods. Model compression and performance evaluation were conducted on a single RTX 3090 24GB GPU with $224\times 224$ resolution. Our approach does not require any fine-tuning process and compresses models by adjusting the decomposition ranks. The decomposition ranks were configured in a layer-wise hierarchical manner. All ablation experiments are based on the official implementation of COMCAT (Xiao et al., 2023), and other SVD methods are based on the official implementation of FLAR-SVD (Thoma et al., 2025). For a fair comparison, all decomposition methods start from the same checkpoints.

### 4.2 COMPARISON WITH PREVIOUS METHODS

Our method is designed for attention decomposition in Transformer-based vision models. In Table 1, we conduct experiments where our approach is integrated into several existing decomposition methods in order to evaluate its effectiveness in diverse settings. In this configuration, no activation information is incorporated, which makes our method relatively simple. For comparison, we applied our method to the following: FWSVD, ASVD and SVD-LLM, which were originally introduced for LLMs; and FLAR-SVD, which was proposed for vision models. Experimental results demonstrate that our attention decomposition approach consistently improves both accuracy and latency across all baselines. Notably, when applied to ASVD, our method shows substantial gains of 8.5% in accuracy and 7.7% in latency on DeiT-Small. Furthermore, our best-performing baseline on DeiT-Base achieves significant compression benefits, reducing parameters by 43.0%, GFLOPs by 43.3%, and latency by 23.5%, while resulting only a 2.5% drop in accuracy. Therefore, these results highlight the effectiveness and generality of our approach for attention decomposition. Since the attention module is a critical bottleneck for latency, the improvements observed in this component have a particularly pronounced impact on overall model efficiency.

### 4.3 EFFECTIVENESS OF OUR UNILATERAL WEIGHT DECOMPOSITION

To evaluate the effectiveness of our approach, we compare the proposed unilateral weight decomposition with the per-weight decomposition and the combined weight decomposition (Xiao et al., 2023), as illustrated in Figure 1(a) and (b). Since our method explores the effect of the weight decomposition in the attention mechanism, we conduct experiments by varying the ranks in the

| Method | DeiT-Small (79.9%) | | | DeiT-S distill (80.9%) | | | DeiT-Base (81.8%) | | |
|---|---|---|---|---|---|---|---|---|---|
| # of Params. in MHA ↓ | 20% | 40% | 60% | 20% | 40% | 60% | 20% | 40% | 60% |
| Per-Weight Decomposition | 75.7 | 67.2 | 31.4 | 73.0 | 56.2 | 23.0 | 80.7 | 79.0 | 71.4 |
| Combined Weight Decomposition | 78.3 | 74.2 | 54.2 | 79.8 | 76.1 | 63.6 | 81.6 | 79.4 | 65.7 |
| Unilateral Weight Decomposition | **78.4** | **74.4** | **65.4** | **79.9** | **76.5** | **69.3** | **81.7** | **80.4** | **72.1** |

| Method | DINOv2 | | | DeiT3-Large | | |
|---|---|---|---|---|---|---|
| | Params in MHA | Top-1 ↑ | Latency ↓ | Params in MHA | Top-1 ↑ | Latency ↓ |
| Baseline | 28.3 | 81.2 | 38.6 | 100.7 | 86.8 | 99.1 |
| Per-Weight Decomposition | 22.5 | 65.5 | 37.3 | 50.3 | 82.3 | 91.7 |
| Combined Weight Decomposition | 22.5 | 64.2 | 36.0 | 50.3 | 81.7 | 84.5 |
| Unilateral Weight Decomposition | 22.5 | **73.2** | 35.7 | 50.3 | **83.0** | 81.9 |

| Method | ViT-Large | | | Swin-Large | | |
|---|---|---|---|---|---|---|
| | Params in MHA | Top-1 ↑ | Latency ↓ | Params in MHA | Top-1 ↑ | Latency ↓ |
| Baseline | 100.7 | 84.3 | 100.6 | 62.8 | 86.3 | 67.8 |
| Per-Weight Decomposition | 50.3 | 75.0 | 91.4 | 31.4 | 80.4 | 63.9 |
| Combined Weight Decomposition | 50.3 | 74.9 | 83.6 | 31.4 | 79.2 | 58.3 |
| Unilateral Weight Decomposition | 50.3 | **77.7** | 81.1 | 31.4 | **81.2** | 57.3 |

Table 2: Comparison of unilateral weight decomposition with the per-weight decomposition and the combined weight decomposition on ImageNet-1K. The results of the first table are reported in terms of Top-1 Accuracy under different parameter reduction ratios in multi-head attention (MHA). The results of the second table are reported with various vision models.

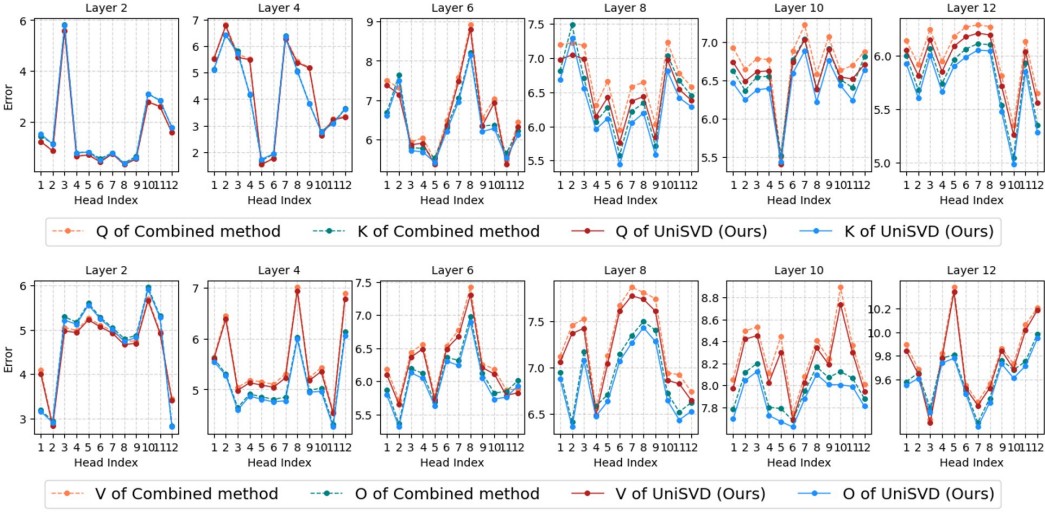

Figure 3: Error comparison with the combined weight decomposition method and our proposed method across layers. Results are measured on the DeiT-Base model with a reduced rank of 24.

MHA layers to set different levels of the parameter reduction in Table 2. The results consistently demonstrate that our unilateral decomposition achieves higher performance across all parameter reduction settings. In particular, under a large reduction ratio of 60%, our method significantly outperforms the combined weight decomposition with improvements of 11.2%, 5.7%, and 6.4% for DeiT-Small, DeiT-Small distilled and DeiT-Base, respectively. Moreover, for DeiT-Base, even when the parameter count is reduced by 20%, the resulting accuracy drop remains extremely small, only about 0.1%, which highlights the stability of our approach under light compression as well. In Table 2, we also evaluate on various vision models, including large models. Our method consistently surpasses two other methods on the large vision models. This demonstrate that UniSVD can generalize to diverse attention-based architectures. These results indicate that decomposing only unilateral weight pairs ($W_i^Q$-$W_i^K$, $W_i^V$-$W_i^O$) effectively mitigates information loss from the original weights, thus making our method a robust and efficient decomposition strategy for attention-based models.

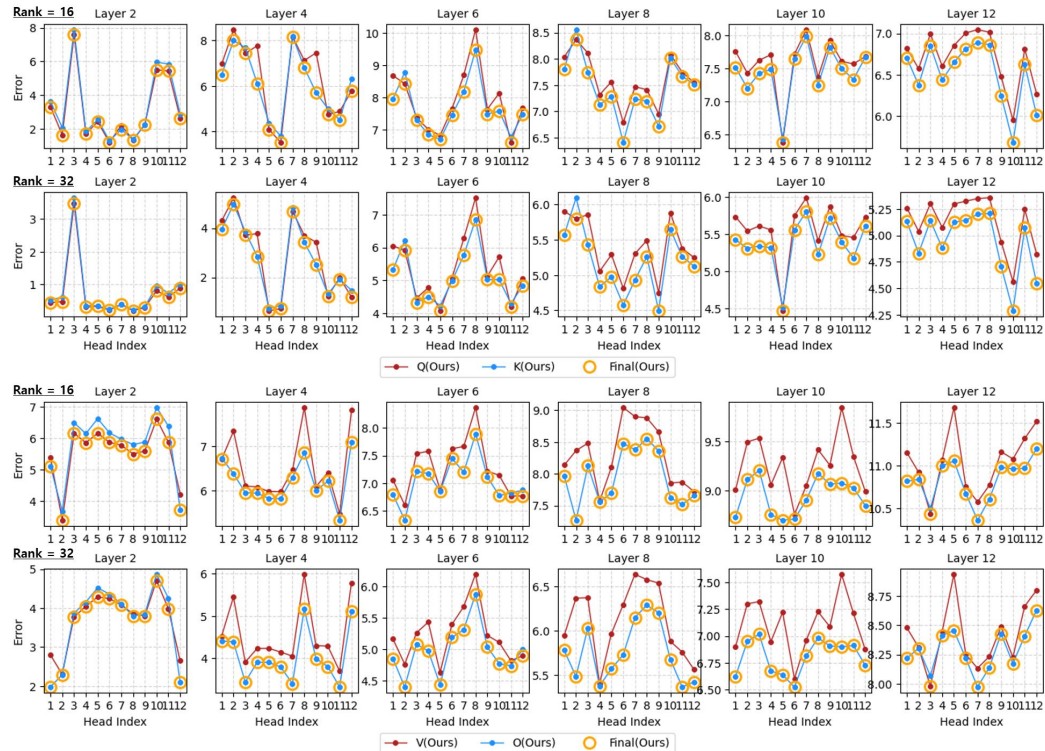

Figure 4: Visualization of head-wise dynamic selection across layers. Results are shown for reduced ranks 16 and 32 on the DeiT-Base model.

## 4.4 ERROR COMPARISON WITH UNISVD AND COMBINED METHOD

Our UniSVD evaluates the approximation error in the individual space of each component $W^Q$, $W^K$, $W^V$, and $W^O$. Accordingly, the low-rank approximation error in our method can be formulated as follows:

$$L_I^A(r) = ||W^A - \text{SVD}(W^A)||_F, \qquad L_I^B(r) = ||W^B - \text{SVD}(W^B)||_F, \tag{11}$$

where $W^A$ and $W^B$ denote weight matrices corresponding to either $(W^Q, W^{K^\top})$ or $(W^V, W^O)$, and $r$ represents the reduced rank used for approximation. $L_I^A$ and $L_I^B$ denote the error functions for the individual weights $W^A$ and $W^B$, respectively. For comparison with the combined method, we multiply the low-rank approximated combined weight by the inverse of either $W^A$ or $W^B$, and then evaluate the error in the corresponding individual space, as formulated below:

$$
\begin{aligned}
L_C^A(r) &= ||W^A - \text{SVD}(W^A W^B) W^{B^{-1}}||_F, \\
L_C^B(r) &= ||W^B - W^{A^{-1}} \text{SVD}(W^A W^B))||_F,
\end{aligned} \tag{12}
$$

where $L_C^A(r)$ and $L_C^B(r)$ denote the error functions of the combined weight, transformed into the individual spaces of $W^A$ and $W^B$, respectively. As shown in Figure 3, we visualize $L_I$ and $L_C$ at rank of 24 on the DeiT-Base model. The experimental results show that in the early layers, the error gap between our UniSVD and the combined method is negligible. However, as the depth increases, the combined method tends to exhibit larger errors compared to the individually low-rank approximated weights. These findings, summarized in Table 2 and Figure 3, demonstrate that the dynamically selected unilateral SVD provides a more effective decomposition strategy than the combined method. We additionally conducted error and performance comparisons in the combined space, with the corresponding analysis reported in Appendix A. Furthermore, the error analysis in the individual space across all layers is provided in Appendix B.

| Method | Selected Elements | | | | DeiT-Small | | | DeiT-S distilled | | | DeiT-Base | | |
|---|---|---|---|---|---|---|---|---|---|---|---|---|---|
| | $Q$ | $K$ | $V$ | $O$ | 20% | 40% | 60% | 20% | 40% | 60% | 20% | 40% | 60% |
| Fixed Selection | ✓ | | ✓ | | 75.4 | 65.9 | 39.7 | 78.1 | 68.9 | 40.1 | 81.4 | 79.5 | 61.8 |
| | ✓ | | | ✓ | 78.2 | 69.0 | 59.0 | 79.4 | 70.4 | 59.3 | 81.6 | 79.4 | 70.3 |
| | | ✓ | ✓ | | 75.8 | 71.3 | 46.5 | 78.8 | 75.8 | 52.7 | 81.5 | 80.4 | 65.6 |
| | | ✓ | | ✓ | 78.3 | 74.2 | 65.3 | 79.6 | 76.1 | 69.3 | 81.7 | 80.3 | 71.7 |
| Dynamic Selection | | | | | **78.4** | **74.4** | **65.4** | **79.9** | **76.5** | **69.3** | **81.7** | **80.4** | **72.1** |

Table 3: Ablation study comparing fixed selection cases and the dynamic selection method for choosing less rank-sensitive elements within the $Q$-$K$ and $V$-$O$ pairs. Results are reported on DeiT-Small, DeiT-Small distilled, and DeiT-Base under different parameter reduction ratios. More experiments and visualizations are provided in Appendix.

| Method | Object Detection | | | Semantic Segmentation | | |
|---|---|---|---|---|---|---|
| | Params (M) | GFLOPs | mAP | Params (M) | GFLOPs | mIoU |
| Baseline | 195.2 | 756.9 | 57.2 | 305.5 | 959.6 | 77.4 |
| Combined Weight Decomposition | 184.6 | 680.4 | 54.1 | 280.3 | 836.3 | 76.8 |
| Unilateral Weight Decomposition | 184.6 | 680.4 | **55.4** | 280.3 | 836.3 | **76.9** |

Table 4: Comparison with the combined weight decomposition method and our UniSVD on downstream tasks. DINO is adopted as baseline for object detection. SETR is adopted as baseline for semantic segmentation.

## 4.5 ANALYSIS OF HEAD-WISE DYNAMIC SELECTION

We employ a head-wise dynamic selection strategy for the $W^Q$-$W^K$ and $W^V$-$W^O$ pairs, where the side with the lower low-rank approximation error is chosen for decomposition. This design allows the decomposition process to adapt at the head level, ensuring that the most stable component in terms of approximation error is selected. To further analyze the detailed selection behavior, we visualize the head-wise errors across all layers, as shown in Figure 4. This analysis is conducted on the DeiT-Base model, where $W^Q$, $W^K$, and $W^V$ are in $\mathbb{R}^{C \times d_h}$ and $W^O$ is in $\mathbb{R}^{d_h \times C}$, with the embedding dimension $C = 768$ and the head dimension $d_h = 64$. Thus, the full-rank size of each weight matrix is 64, and we visualize the approximation behavior at reduced ranks of 16 and 32. The results show that in the $W^Q$-$W^K$ pair, $W^K$ is more often selected, while in the $W^V$-$W^O$ pair, $W^O$ is more frequently chosen. This indicates that $W^K$ and $W^O$ exhibit lower sensitivity to low-rank approximation. Moreover, in the early layers (e.g., layer 2), $W^Q$ and $W^K$ tend to be selected more frequently, whereas in the middle-to-late layers the selection shifts primarily toward $W^K$ and $W^O$. In addition, we provide the analysis results for the full set of layers in Appendix D.

## 4.6 EFFECTIVENESS OF DYNAMIC SELECTION

Our UniSVD dynamically selects the weights of each $W^Q$-$W^K$ and $W^V$-$W^O$ pair based on their sensitivity to low-rank approximation across heads and layers. Rather than enforcing a uniform decomposition rule, this strategy adaptively determines which side of the pair should be factorized, thereby ensuring that the more stable component with respect to approximation error is preserved. To verify its effectiveness, we conducted experiments by fixing the decomposition choice for all four possible configurations of $Q$-$K$ and $V$-$O$ selection, and compared these cases with our dynamic selection strategy, as shown in Table 3. The results demonstrate that our method more effectively prevents information loss in the weights. As the reduction rate increases, decomposing $W^K$ and $W^O$ yields performance that closely approaches that of dynamic selection, indicating that the low-rank approximation error of $W^K$ and $W^O$ becomes smaller at higher reduction rates. Furthermore, when compared with the least efficient fixed selection case (*i.e.*, decomposing $W^Q$ and $W^V$), our dynamic approach achieves performance improvements of 3.0%, 1.8% and 0.3% at a 20% parameter reduction on DeiT-Small, DeiT-Small distilled, and DeiT-Base, respectively.

| Method | EVA | | | EVA-CLIP | | | | | | | | |
|---|---|---|---|---|---|---|---|---|---|---|---|---|
| | | | | | | T2I Retrieval | | | I2T Retrieval | | | |
| | Params (B) | GFLOPs | Top-1 | Params (M) | GFLOPs | R@1 | R@5 | R@10 | R@1 | R@5 | R@10 | |
| Baseline | 1.0 | 620.6 | 89.6 | 1136.6 | 299.9 | 74.6 | 92.3 | 95.2 | 90.0 | 98.6 | 99.4 | |
| Combined Weight Decomposition | 0.9 | 550.5 | 86.5 | 1021.2 | 267.6 | 73.7 | 91.3 | 95.1 | 88.3 | 98.3 | 99.4 | |
| Unilateral Weight Decomposition | 0.9 | 550.5 | **87.0** | 1021.2 | 267.6 | **74.0** | **91.5** | **95.1** | **89.3** | **98.3** | **99.4** | |

| Method | LLaVA1.5-13B | | | | | | |
|---|---|---|---|---|---|---|---|
| | Params (M) of Vision Tower | GFLOPs | VQAv2 | GQA | TextVQA | MMBench | MME |
| Baseline | 303.5 | 191.1 | 78.5 | 61.9 | 58.2 | 64.6 | 1504.6 |
| Combined Weight Decomposition | 272.0 | 167.8 | 73.6 | 59.4 | 53.1 | 63.0 | 1374.7 |
| Unilateral Weight Decomposition | 272.0 | 167.8 | **74.7** | **59.8** | **53.1** | **63.0** | **1408.0** |

Table 5: Comparison with the combined weight decomposition method and our UniSVD on vision-language models. EVA is evaluated on ImageNet-1K. EVA-CLIP is evaluated on Flickr30k. LLaVA1.5 is evaluated on widely-used five benchmarks.

## 4.7 GENERALIZABILITY OF UNISVD

To validate the generalization ability, we experimented on downstream tasks, *i.e.*, object detection and semantic segmentation. We applied our method and the combined weight decomposition method to DINO, which uses Swin Transformer, for object detection, and SETR, which uses ViT, for semantic segmentation. In Table 4, the results indicate that our method is more effective than the combined weight decomposition method on both downstream tasks, with a particularly 1.3% higher performance on the object detection. In Table 5, we also experimented with vision-language models for further evaluation. Our method showed better performance on various vision-language tasks. These results demonstrate the generalization ability of our UniSVD.

## 5 CONCLUSION

In this paper, we proposed a novel and efficient method, Unilateral Singular Value Decomposition (UniSVD), which applies low-rank approximation by decomposing only one side of each weight pair ($W^Q$-$W^K$ and $W^V$-$W^O$) in the Multi-Head Attention of Transformer-based vision models. UniSVD dynamically selected the weights to decompose based on their sensitivity to low-rank approximation across heads and layers, thereby better preserving information contained in rank-sensitive weights. Our approach overcomes the inefficiency of per-weight decomposition methods and achieves better performance retention than combined decomposition methods, especially in low-rank approximation. Experimental results demonstrated that our method effectively minimizes information loss compared to the per-weight decomposition and the combined decomposition approaches. Furthermore, UniSVD can be universally integrated into the attention decomposition part of various decomposition methods, consistently achieving significant performance improvements and latency reductions.

**Discussions and Future Works.** Our method is designed as a decomposition technique for the multi-head attention module in Transformer-based vision models. Accordingly, it demonstrates substantial efficiency improvements in terms of computational complexity and latency, which are particularly dependent on the attention mechanism. However, it is not yet straightforward to directly apply our method to MLP components, where parameter reduction may yield greater benefits. For large language models, our method can be directly applied to $V$-$O$ weights, whereas $Q$-$K$ weights require consideration of rotary position embeddings. We believe our approach can be effectively extended to LLMs with suitable adaptations, and we leave this extension for future work.

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

# A   ERROR COMPARISON WITH UNISVD AND COMBINED METHOD IN COMBINED SPACE

To compare the combined and UniSVD decomposition methods, we additionally evaluated errors and validated performance in the combined space (i.e., $W^A W^B$). In this setting, the error functions for both the individual weights and the combined weights are defined as follows:

$$L_I^A(r) = ||W^A W^B - SVD(W^A)W^B||_F, \quad L_I^B(r) = ||W^A W^B - W^A SVD(W^B)||_F, \quad (13)$$

$$L_C^{AB}(r) = ||W^A W^B - SVD(W^A W^B)||_F, \quad (14)$$

Here, $W^A$ and $W^B$ denote weight matrices that can correspond to either $(W^Q, W^{K\top})$ or $(W^V, W^O)$, and $r$ represents the reduced rank used for approximation. The error function $L_C^{AB}$ measures the approximation error of the combined weight, whereas $L_I^A$ and $L_I^B$ evaluate the error under the individual weights (e.g. $W^A, W^B$) decomposition adopted in our UniSVD. In practice, UniSVD selects the decomposition side by comparing $L_I^A$ and $L_I^B$, choosing the one with the lower error. As shown in Figure 5, we visualize $L_I^A, L_I^B$ and $L_C^{AB}$ at rank of 24 on the DeiT-Base model. Although $L_C^{AB}$ is mathematically guaranteed, and empirically observed in Figure 5, to yield lower approximation error than either $L_I^A$ or $L_I^B$, Table 6 demonstrates that UniSVD with selective individual decomposition achieves superior performance. These experimental results demonstrate that optimizing the error based on the combined weight does not align well with the actual performance. Therefore, searching for the optimal decomposition in the individual spaces rather than in the combined space is more effective in enhancing performance. Therefore, our method evaluates the approximation error within the individual space.

# B   ADDITIONAL ERROR COMPARISON WITH UNISVD AND COMBINED METHOD IN INDIVIDUAL SPACE

As presented in Figure 3 of the main paper, we initially provided a limited visualization focusing on a subset of layers. To complement this, we include in Figure 7 the error comparison results in the individual space across all layers. The extended analysis confirms that our proposed method consistently achieves lower errors than the combined method, and these findings are well aligned with the experimental results reported in Table 6.

# C   ADDITIONAL ANALYSIS OF SINGULAR VALUE DISTRIBUTIONS

As discussed in Figure 2 of the main paper, we initially presented a limited visualization of the singular value distributions for selected layers and heads. To provide a more comprehensive analysis, we extend this study by including additional results across all heads and layers. As shown in Figures 8 and 9, the singular values of the combined weights exhibit a sharp increase in the low-rank regime compared to those of the individual weights.

# D   ADDITIONAL ANALYSIS OF HEAD-WISE DYNAMIC SELECTION

As presented in Figure 4 of the main paper, we initially provided a limited visualization focusing on a subset of layers. To complement this, we extend the analysis by including additional results for a broader range of layers and heads in Figures 10 and 11. Consistent with the findings in Figure 4, the results indicate that within the $W^Q$-$W^K$ pair, $W^K$ is selected more frequently, whereas within the $W^V$-$W^O$ pair, $W^O$ tends to be chosen more often. Furthermore, in the early layers, the selection is biased toward $W^Q$ and $W^K$, while in the middle-to-late layers the preference shifts predominantly to $W^K$ and $W^O$.

# E   PSEUDOCODE OF UNILARTERAL SINGULAR VALUE DECOMPOSITION

We present a PyTorch-like pseudocode of the unilateral weight decomposition in Algorithm 1. This pseudocode is provided to illustrate the step-by-step procedure of our method, showing how the dynamic selection and decomposition operations can be implemented in practice.

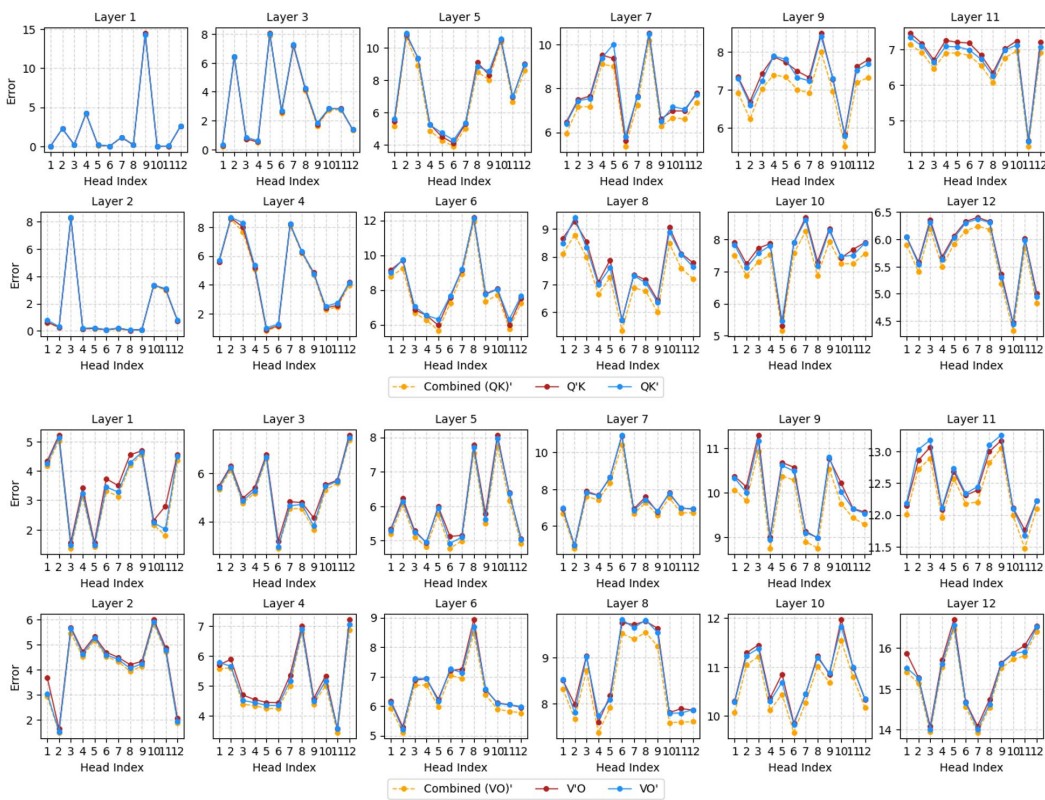

Figure 5: Error comparison in combined space with the combined weight decomposition method and our proposed method across layers. Results are measured on the DeiT-Base model with a reduced rank of 24.

| Method | Reduced Rank | #Param Reduction in MHA (%) | Top-1 |
|---|---|---|---|
| Combined Weight Decomposition | 16 | 75 | 19.1 |
| Unilateral Weight Decomposition | 16 | 75 | **41.9** |
| Combined Weight Decomposition | 24 | 62.5 | 63.6 |
| Unilateral Weight Decomposition | 24 | 62.5 | **69.5** |

Table 6: Comparison of the unilateral and combined weight decomposition at rank 16 and 24 on DeiT-Base.

## F    STATEMENT ON THE USE OF LARGE LANGUAGE MODELS

In the interest of transparency and in compliance with the ICLR 2026 guidelines, we report that a large language model (LLM) was used to assist in the refinement of this manuscript.

**Scope of Use.**    The LLM was utilized solely as a writing assistant. Its involvement was restricted to the following tasks:

- Correcting grammar, spelling, and punctuation errors.
- Enhancing sentence structure and readability.
- Refining word choices for improved clarity and conciseness.

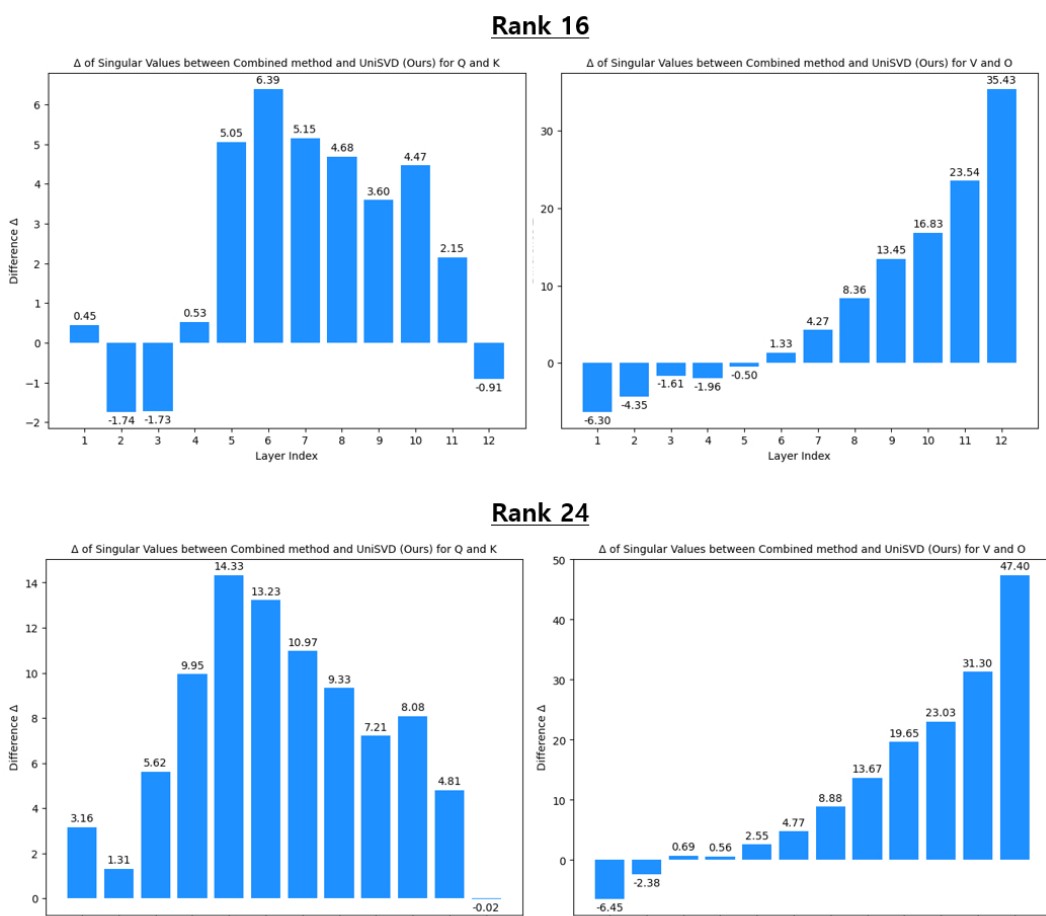

Figure 6: Differences of singular values between the combined weight decomposition method and our unilateral weight decomposition method. Results are measured on a DeiT-Base model.

| Method | Q | K | V | O |
|--------|-----|-----|-----|-----|
| DeiT-Base | 20% | 80% | 10% | 90% |
| DINOv2 | 15% | 85% | 11% | 88% |
| ViT-Large | 30% | 70% | 7% | 93% |
| DeiT-Large | 35% | 65% | 34% | 66% |
| Swin-Large | 17% | 83% | 30% | 70% |

Table 7: The ratio of Q/K selection and V/O selection for various models.

| Method | DINOv2 | ViT-Large | DeiT-Large | Swin-Large |
|--------|--------|-----------|------------|------------|
| w/o selection (sec) | 0.6 | 1.3 | 1.1 | 1.3 |
| w/ selection (sec) | 1.2 | 2.6 | 2.4 | 2.6 |

Table 8: Wall-clock time comparison for the dynamic selection. Each time is the cumulative time for all layers. Results are measured on a single 3090 GPU.

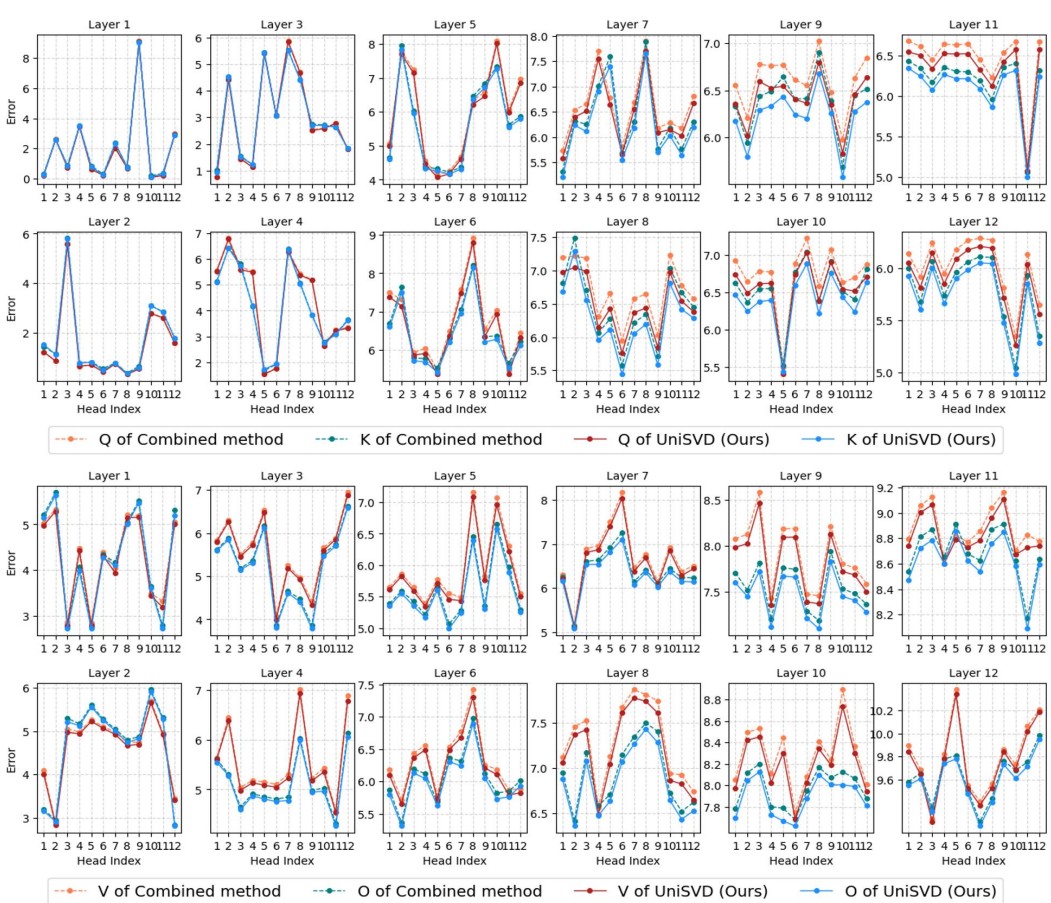

Figure 7: Error comparison with the combined weight decomposition method and our proposed method across all layers. Results are measured on the DeiT-Base model with a reduced rank of 24.

| Metric | DeiT-Base (Base head dim: 64) | | | DeiT-Large (Base head dim: 64) | | |
|---|---|---|---|---|---|---|
| | Rank 16 | Rank 32 | Rank 48 | Rank 16 | Rank 32 | Rank 48 |
| Activation-aware | 36.9 | 77.1 | 81.4 | 52.1 | **83.6** | **86.3** |
| Spectral norm | 37.6 | 77.3 | 81.4 | **59.9** | 83.3 | 84.9 |
| Frobenius norm | **43.2** | **77.5** | **81.5** | 56.6 | 83.0 | 85.1 |
| | ViT-Large (Base head dim: 64) | | | Swin-Large (Base head dim: 32) | | |
| | Rank 16 | Rank 32 | Rank 48 | Rank 8 | Rank 16 | Rank 24 |
| Activation-aware | 23.1 | 75.9 | 80.8 | 21.6 | 81.1 | **85.1** |
| Spectral norm | **33.6** | 75.8 | 80.9 | **22.4** | **81.5** | 84.8 |
| Frobenius norm | 30.0 | **77.7** | **81.2** | **22.4** | 81.2 | 84.9 |

Table 9: Comparison with other error metric method on various models.

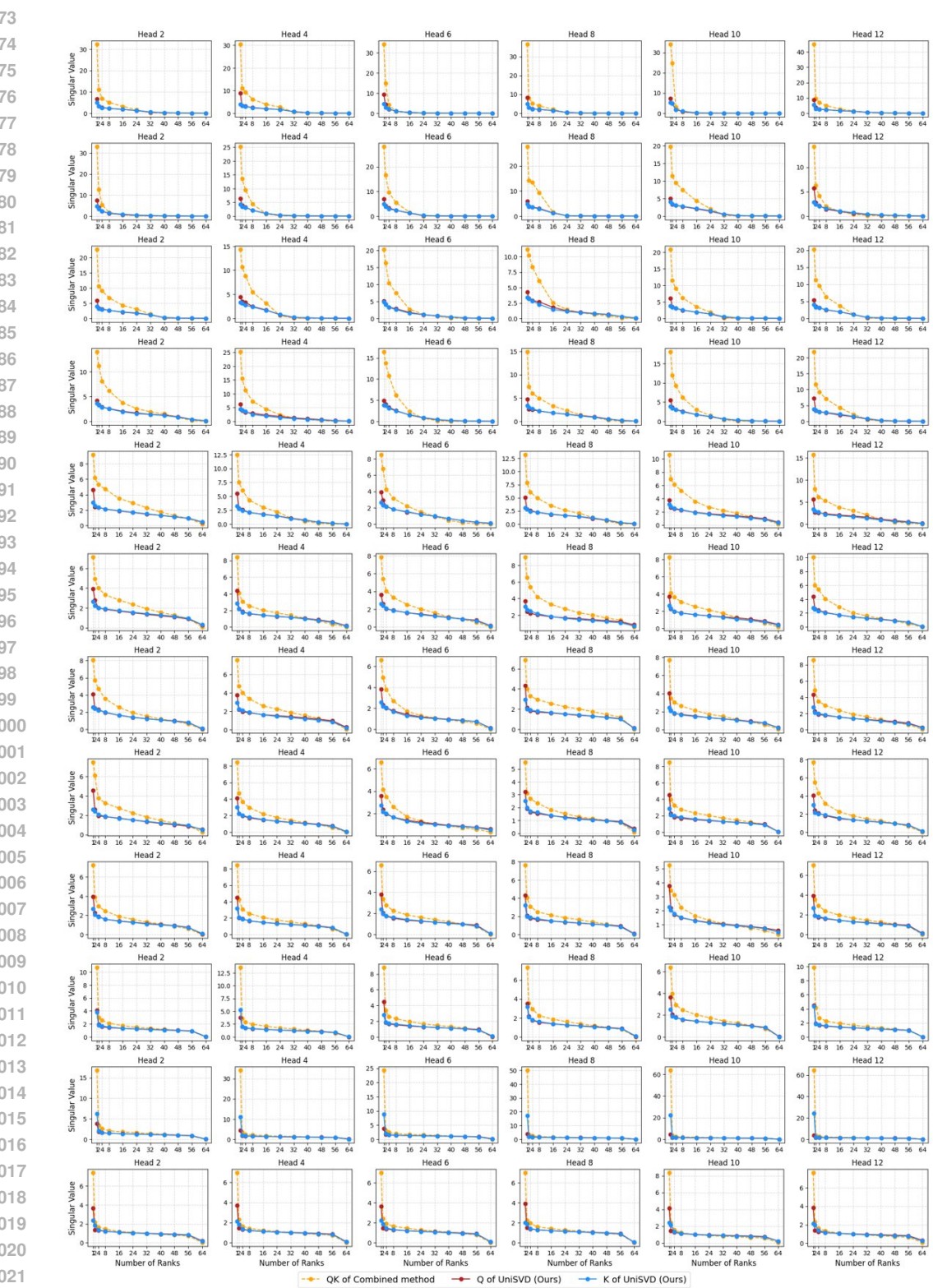

Figure 8: The singular value distributions of Q-K at each rank for the combined weight matrices and the individual weight matrices.

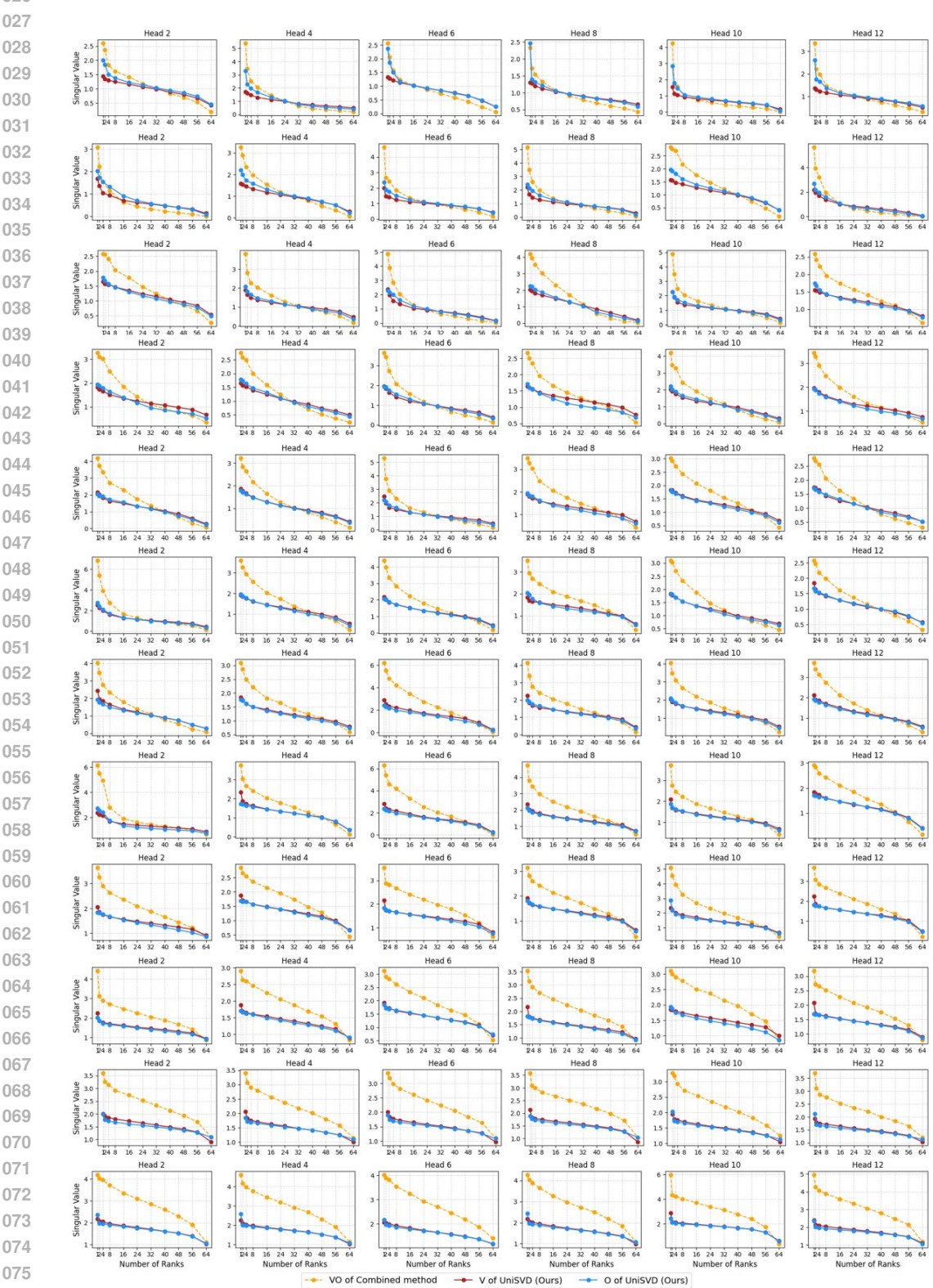

Figure 9: The singular value distributions of V-O at each rank for the combined weight matrices and the individual weight matrices.

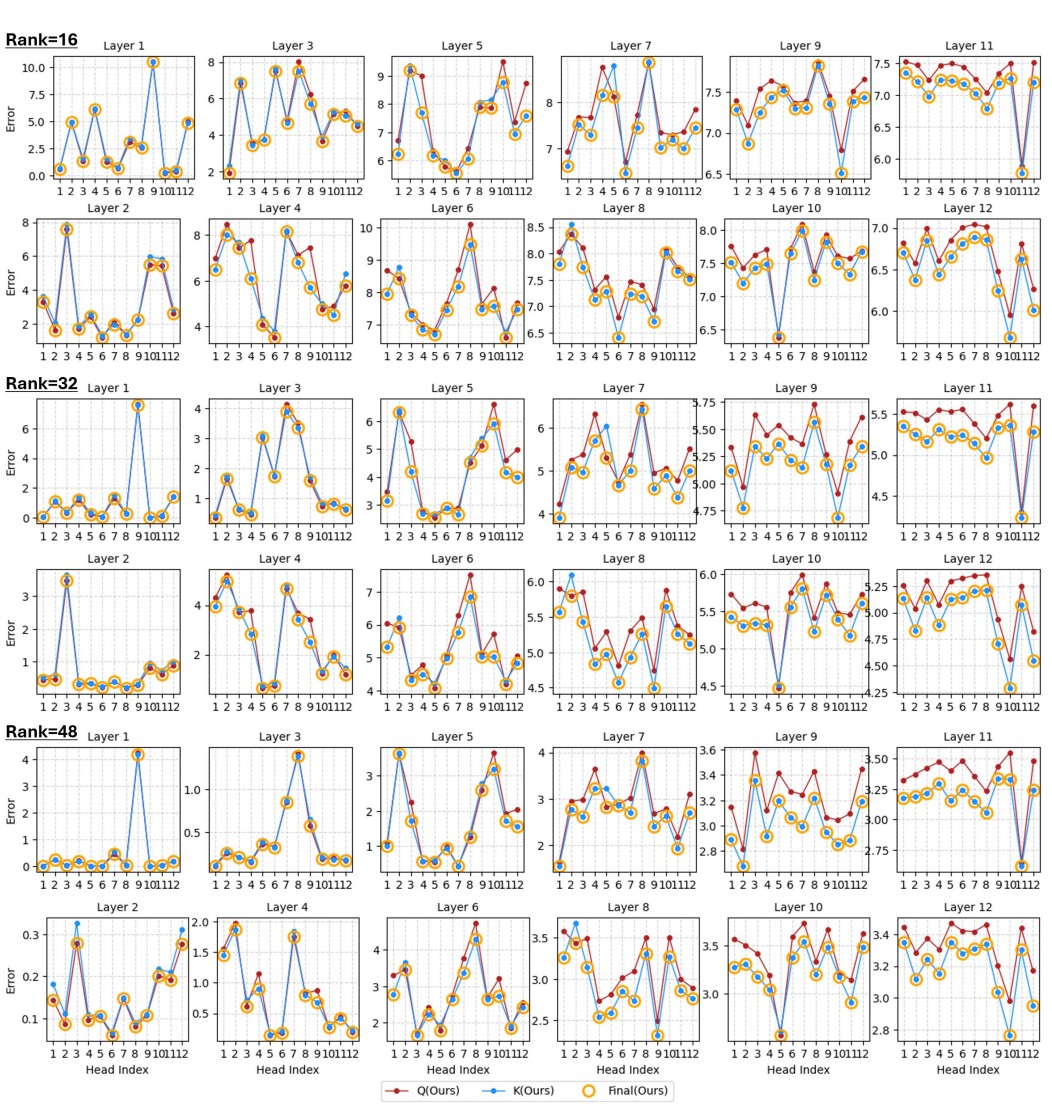

Figure 10: Additional visualization of head-wise dynamic selection within the $W^Q$-$W^K$ pair.

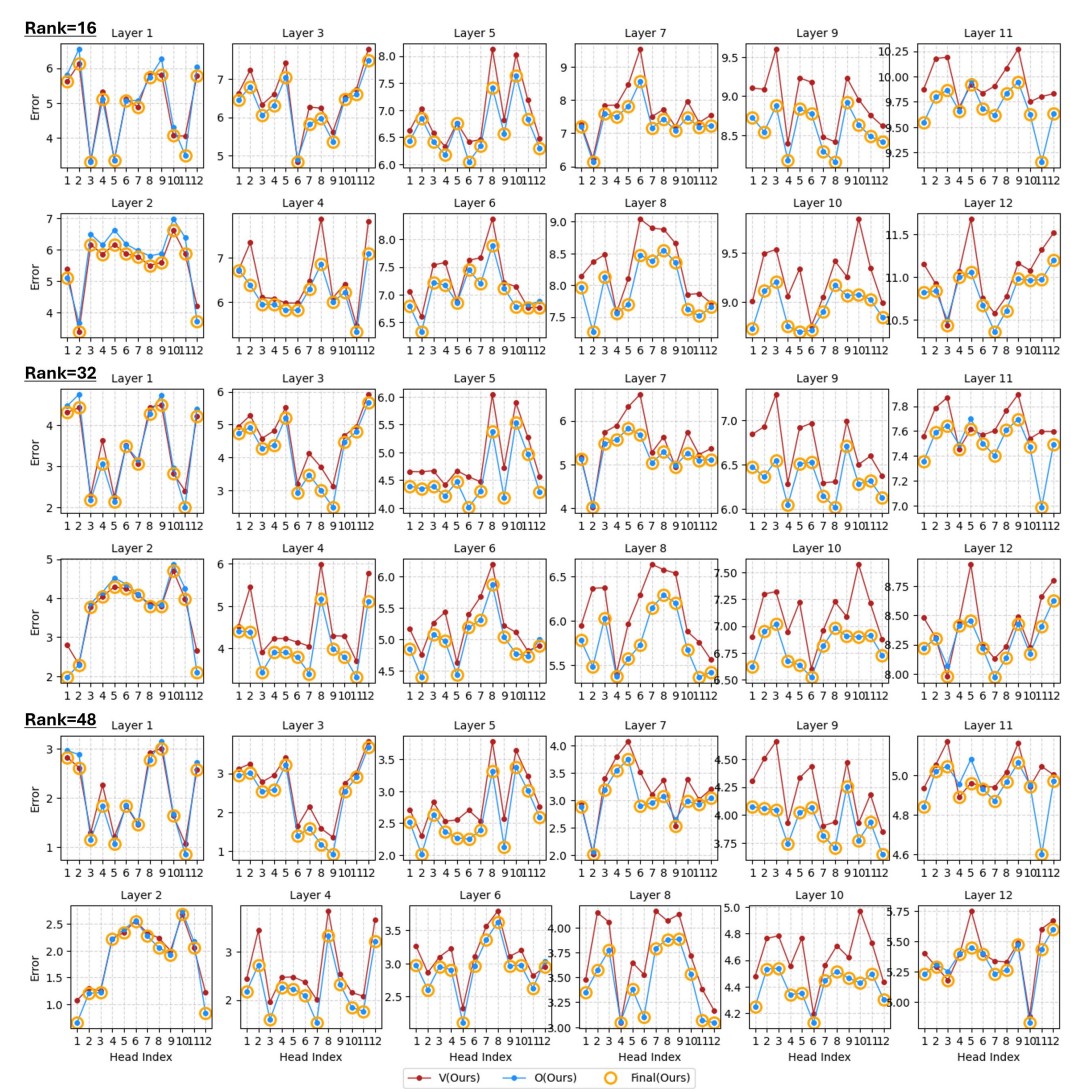

Figure 11: Additional visualization of head-wise dynamic selection within the $W^V$-$W^O$ pair.

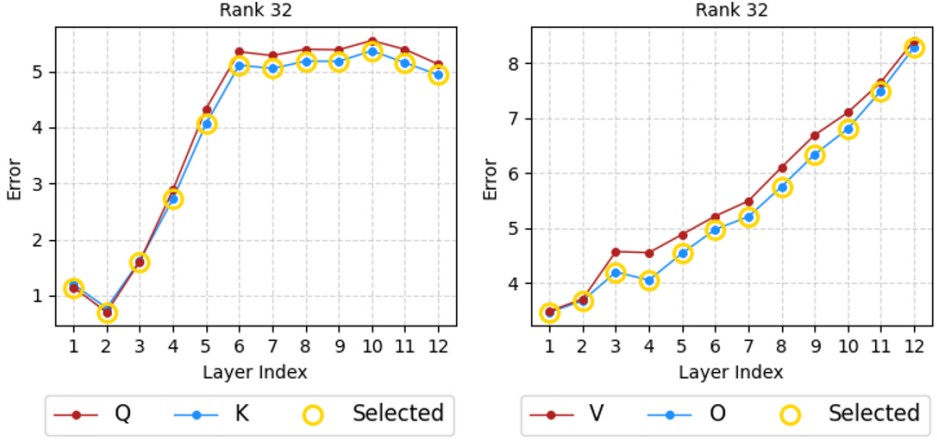

Figure 12: Additional visualization of layer-wise dynamic selection.

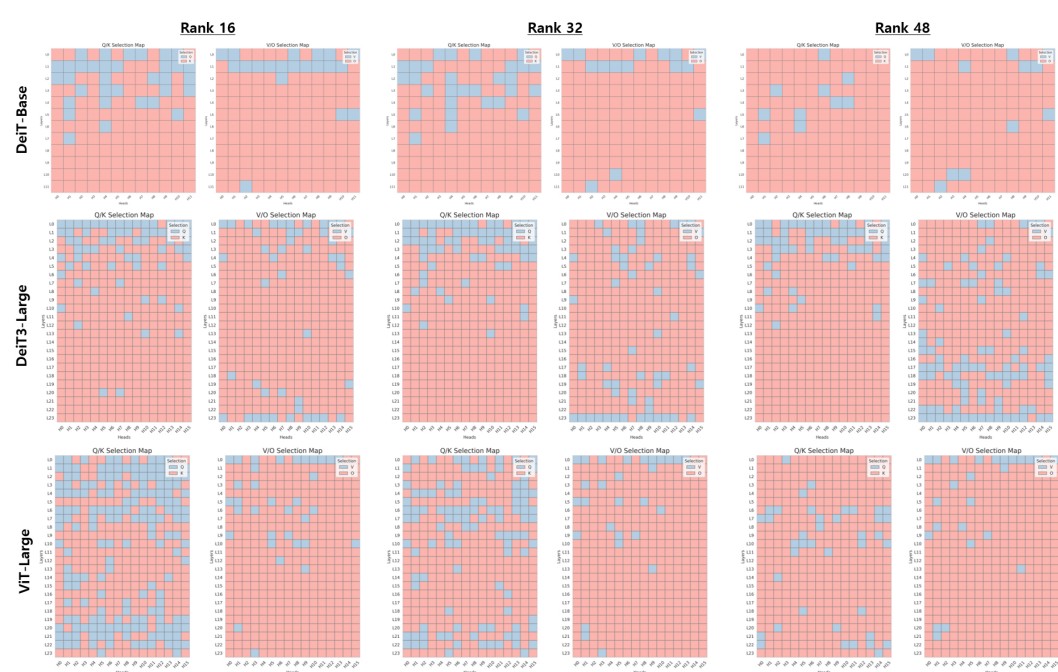

Figure 13: Additional visualization of selection maps per-layer/head on various models and ranks.

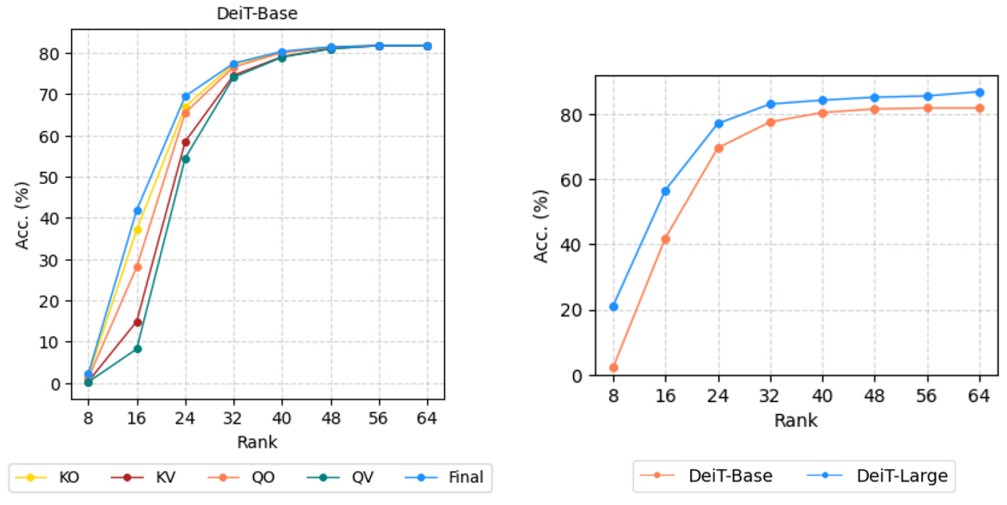

Figure 14: (left) Comparison of the sensitivity to rank. The dynamic selection show better robustness compared to the fixed cases. (right) Rank-Performance curves on DeiT-Base and DeiT-Large.

**Algorithm 1:** UniSVD: PyTorch-like pseudocode

```
def UniSVD(W1, W2, rank, transpose):
   # SVD for calculating loss
   U1, S1, V1 = torch.linalg.svd(W1.detach(),
    full_matrices=False)
   U2, S2, V2 = torch.linalg.svd(W2.detach(),
    full_matrices=False)

   U1 = U1[:, :rank]
   S1 = S1[:rank]
   V1 = V1[:rank, :]
   Wr1 = U1 * S1 @ V1

   U2 = U2[:, :rank]
   S2 = S2[:rank]
   V2 = V2[:rank, :]
   Wr2 = U2 * S2 @ V2

   # Frobenius norm-based loss
   loss1 = torch.norm(W1 - Wr1)
   loss2 = torch.norm(W2 - Wr2)

   # Select less rank-sensitive weight
   if loss1 < loss2 :
     # Decompose only W1
     A = V1
     if transpose:  # For Q-K pair
        B = (U1 * S1).transpose(-1, -2)
        B = B @ W2
     else:    # For V-O pair
        B = W2 @ (U1 * S1)
   else:
     # Decompose only W2
     if transpose:    # For Q-K pair
        A = (U2 * S2).transpose(-1, -2)
        A = A @ W1
        B = V2
     else:    # For V-O pair
        A = V2 @ W1
        B = U2 * S2

   return A, B
```

