# OpenReview forum: "UniSVD: Unilateral Weight Decomposition for Attention-based Vision Models"
_ICLR.cc/2026/Conference — Submitted to ICLR 2026_

### Official Review · Reviewer_Rygs · 2025-10-27

**Soundness:** 3
**Presentation:** 4
**Contribution:** 3
**Rating:** 8
**Confidence:** 2

**Summary:**

The key idea is that to decompose only one side of (Q,K) and (V,O) per head, chosen by rank-sensitivity, to improve the accuracy–efficiency trade-off vs per-weight and combined decompositions and positions against per-weight SVD inefficiency and COMCAT’s combined scheme.

The paper claims broad plug-in gains across FWSVD, ASVD, SVD-LLM, FLAR-SVD without fine-tuning. This is backed up by some impressive results - notably in table 2 - beats per-weight and combined methods across 20/40/60% MHA reduction (Table 2); large gains at 60%.

Error analyses across layers support the mechanism

**Strengths:**

Simple idea which results in some good results.

The idea is simple, training-free, and plugs into multiple SVD families, improving both Top-1 and latency without fine-tuning. Under heavy MHA reduction, unilateral consistently outperforms combined by large margins. I think this is a nice result which is presented in a very clear way.

**Weaknesses:**

End-to-end efficiency reporting -  provide VRAM and throughput across batch/sequence lengths (besides latency) for each baseline + UniSVD; include wall-clock for the selection pass.
The following would improve the paper I feel

Compare Frobenius-error selection with alternatives (spectral norm, activation-aware criteria like ASVD/SVD-LLM whitening) to confirm robustness.

Add ViT-L/16 or Swin-T, and a second dataset to show generality beyond DeiT/ImageNet-1K.

Granularity of ranks. Release per-layer/head ranks and selection maps; add sensitivity curves (Top-1 vs rank) to guide practitioners.

Theory–performance link. The appendix notes combined-space error may not track accuracy; expand analysis on why individual-space optimisation aligns better with performance.

Ablations on latency sources. Separate gains from fewer sublayers vs cache/matmul effects to clarify where the speedup comes from.

**Questions:**

Can you provide full VRAM/throughput and batch/sequence sweeps, and quantify the one-off cost of computing head-wise SVD errors for selection?

Do you have results on non-DeiT architectures or another dataset to confirm portability?

Could unilateral decomposition extend to MLP blocks (e.g., only decompose one of in/out projections) or to LLM attention with RoPE? Any problems you foresee?

The appendix shows combined-space error isn’t predictive of accuracy. Can you formalise why individual-space optimisation is better aligned, or give counter-examples?

---

> ### Author Response · Authors · 2025-11-21
> **Official Comment for Reviewer Rygs (1/2)**
>
> Thanks for the reviewer Rygs's insightful comments. We addressed concerns about 'Generality of UniSVD' in **[Common Response 1 & 2]**, and concerns about 'Wall-clock for selection pass' in **[Common Response 3]**.
>
> ---
>
> ### **Additional Efficiency Reporting**
> - **DeiT-Base / Sequence length : 197 (Input resolution : 224 x 224)**
> Rank| Batch size | VRAM (MB) | Latency (ms) |
> ----|----|----|-----|
> 64 (Base)|16| 776 | 24.2
> 64 (Base)|64| 1174 |89.3
> 64 (Base)|256| 3004 |353.1
> 32|16| 633| 20.0
> 32|64| 1049|76.8
> 32|256| 2005|302.4
> 16|16| 577|18.5
> 16|64| 857|70.2
> 16|256|1856|275.6
>
> - **DeiT-Base / Sequence length : 577 (Input resolution : 384 x 384)**
> Rank| Batch size | VRAM (MB) | Latency (ms) |
> -----|-----|------|------|
> 64 (Base)|16| 1402| 79.8
> 64 (Base)|64| 4254|305.6
> 64 (Base)|256| 15048|1226.4
> 32|16| 949| 68.0
> 32|64| 2487|269.2
> 32|256| 8606|1079.6
> 16|16| 900|64.1
> 16|64| 2392|243.5
> 16|256|8282|979.6
>
> - **DeiT-Large / Sequence length : 197 (Input resolution : 224 x 224)**
> Rank| Batch size | VRAM (MB) | Latency (ms) |
> ---|---|----|----|
> 64 (Base)|16| 2058| 74.5
> 64 (Base)|64| 2500|302.2
> 64 (Base)|256| 4722|1210.5
> 32|16| 1581| 62.5
> 32|64| 1928|250.6
> 32|256| 3349|1002.3
> 16|16| 1435|55.9
> 16|64| 1758|225.4
> 16|256|3086|891.7
>
> - **DeiT-Large / Sequence length : 577 (Input resolution : 384 x 384)**
> Rank| Batch size | VRAM (MB) | Latency (ms) |
> ---|---|---|---|
> 64 (Base)|16| 2954| 252.8
> 64 (Base)|64| 6298|1014.7
> 64 (Base)|256| 20866|4046.5
> 32|16| 2143| 214.2
> 32|64| 4145|854.2
> 32|256| 12163|3413.1
> 16|16| 1982|190.3
> 16|64| 3916|768.5
> 16|256|11665|3074.0
>
> - **ViT-Large / Sequence length : 197 (Input resolution : 224 x 224)**
> Rank| Batch size | VRAM (MB) | Latency (ms) |
> ----|----|----|----|
> 64 (Base)|16| 2020| 74.2
> 64 (Base)|64| 2400|295.7
> 64 (Base)|256| 4840|1189.0
> 32|16| 1575| 61.9
> 32|64| 1049|250.1
> 32|256| 3345|989.3
> 16|16| 1428|56.0
> 16|64| 1754|221.8
> 16|256|3082|892.9
>
> - **ViT-Large / Sequence length : 577 (Input resolution : 384 x 384)**
> Rank| Batch size | VRAM (MB) | Latency (ms)  |
> -----|-----|------|------|
> 64 (Base)|16| 2934| 252.4
> 64 (Base)|64| 6228|1009.5
> 64 (Base)|256| 20578|4030.3
> 32|16| 2128| 215.0
> 32|64| 4130|859.2
> 32|256| 12149|3436.6
> 16|16| 1965|195.6
> 16|64| 3899|781.3
> 16|256|11647|3126.5
>
> ***Table K. VRAM usage and latency on different batch size and sequence length.***
>
> As suggested by the reviewer, we measured both VRAM usage and latency across several baselines (DeiT-Base, DeiT-Large and ViT-Large). The measurements were conducted under various settings of batch size and sequence length (image resolution), and for different target ranks. In Table K, the results show that VRAM consumption and latency scale proportionally with the sequence length, batch size, and rank. We thank the reviewer for encouraging this more fine-grained analysis, and we would be happy to conduct additional experiments if there are further specific setups the reviewer would like us to consider.
>
> ---
>
> ### **Comparison with Other Error Metric**
> - **DeiT-Base (Base head dim: 64)**
> | Metric     | Rank 16 | Rank 32 | Rank 48 |
> |-----------|--------:|--------:|--------:|
> | Activation| 36.9    | 77.1    | 81.4    |
> | Spectral  | 37.6    | 77.3    | 81.4    |
> | Frobenius | **43.2**| **77.5**| **81.5**|
>
> - **DeiT-Large (Base head dim: 64)**
> | Metric     | Rank 16 | Rank 32 | Rank 48 |
> |-----------|--------:|--------:|--------:|
> | Activation| 52.1    | **83.6**| **86.3**|
> | Spectral  | **59.9**| 83.3    | 84.9    |
> | Frobenius | 56.6    | 83.0    | 85.1    |
>
> - **ViT-Large (Base head dim: 64)**
> | Metric     | Rank 16 | Rank 32 | Rank 48 |
> |-----------|--------:|--------:|--------:|
> | Activation| 23.1    | 75.9    | 80.8    |
> | Spectral  | **33.6**| 75.8    | 80.9    |
> | Frobenius | 30.0    | **77.7**| **81.2**|
>
> - **Swin-Large (Base head dim: 32)**
> | Metric     | Rank 8 | Rank 16 | Rank 24 |
> |-----------|-------:|--------:|--------:|
> | Activation| 21.6   | 81.1    | **85.1**|
> | Spectral  | **22.4**| **81.5**| 84.8   |
> | Frobenius | **22.4**   | 81.2    | 84.9   |
>
> ***Table L. Comparison with other error metric method on various models.***
>
> Following the reviewer’s constructive suggestion, we additionally evaluated alternative error metrics, namely the spectral norm and an activation-aware Frobenius norm, within the proposed UniSVD framework. As summarized in Table L, we measured the performance across different ranks on DeiT-Base, DeiT-Large, ViT-Large, and Swin-Large.
>
> The results show that the best performing metric depends on both the model and the rank. In particular, when the rank is relatively small, the spectral-norm method tends to yield better performance, whereas for higher ranks on DeiT-Large and Swin-Large, the activation-aware Frobenius norm performs slightly better. We believe that identifying which metric is optimal for a given architecture and rank setting remains an interesting direction for future work, and we are grateful for this very constructive suggestion that helps further improve UniSVD within the same framework.

---

> ### Author Response · Authors · 2025-11-21
> **Official Comment for Reviewer Rygs (2/2)**
>
> ### **Per-Head Selection Map and Sensitivity Curve**
>
> To address the reviewer’s request regarding the granularity of ranks, we added rank–performance curves for DeiT-Base and DeiT-Large in **Figure 14 (right)** of the revised draft. In these plots, the rank on the x-axis is set to the same value for all heads, so that the sensitivity of performance to the chosen rank can be clearly observed.
>
> In addition, we also include selection maps for different ranks in **Figure 13** of the revised draft, which visualize whether the Q-K or V-O matrix is selected. These experiments were conducted on DeiT-Base, DeiT-Large, and ViT-Large with ranks set to 16, 32, and 48. The resulting Q/V vs. K/O selections vary across models and ranks, but a consistent trend is observed: the Q/V case tends to be selected more frequently in lower layers, while the K/O case is selected more often overall. We hope that these additional analyses help clarify the rank granularity and selection behavior, and address the reviewer’s concerns.
>
> ---
>
> ### **Counter-Example for the Formalization Perspective**
>
> We thank the reviewer for this careful and insightful comment. Our individual-space optimization strategy provides a consistent empirical explanation for the behavior of the combined method versus our unilateral (UniSVD) method. However, from the perspective of formalizing this, a counter-example arises when comparing per-weight decomposition with combined weight decomposition. As shown in Table 2 of the draft for DeiT-Base with 60% reduction, the per-weight decomposition achieves better performance than the combined method. At the same time, because per-weight decomposition decomposes both sides of Q-K and V-O, it incurs a larger approximation error than the combined method according to the analysis in Figure 3 of our draft.
>
> Therefore, the individual optimization approach yields a counterexample when comparing per-weight decomposition with the combined method. Although the proposed individual optimization approach shows consistent effects for the combined method and our proposed method, it cannot at this stage be considered a formulation that represents all possible cases, and there remains room for further improvement.
> We thank the reviewer for this sharp observation.
>
> ---
>
> ### **Generality Across Diverse Datasets**
> - **Semantic Segmentation**
> | Method    | Param (M)            | GFLOPs             | mIoU  |
> |-----------|------------------|-------------------|-------|
> | SETR (Baseline)      | 305.5            | 959.6             | 77.4 |
> | Combined Weight Decomposition   | 280.3 (↓8.3%)    | 836.3 (↓12.8%)     | 76.8 |
> |  Unilateral  Weight Decomposition  (Ours)  | 280.3 (↓8.3%)    | 836.3 (↓12.8%)     | **76.9** |
>
> ***Table M. Weight decomposition comparison on Cityscapes***
>
> - **Object Detection**
> | Method    | Param (M)            | GFLOPs             | mAP  |
> |-----------|------------------|-------------------|-------|
> | DINO (Baseline)   | 195.2             | 756.9              | 57.2 |
> | Combined Weight Decomposition  | 184.6 (↓5.4%)     | 680.4 (↓10.1%)      | 54.1 |
> | Unilateral  Weight Decomposition (Ours)   | 184.6 (↓5.4%)     | 680.4 (↓10.1%)      | **55.4** |
>
> ***Table N. Weight decomposition comparison on COCO***
>
> In addition to the experiments on other backbones described in the common response, we also conducted experiments on another set of datasets in Table M and N, specifically Cityscapes for semantic segmentation and COCO for object detection. For semantic segmentation on Cityscapes, we applied the proposed method to SETR [1], which uses ViT as its backbone. For object detection on COCO, we used DINO [2] with a Swin Transformer backbone. To specifically assess generality, we applied our method only to the attention layers. The results indicate that our method is more effective than the combined weight decomposition method on both tasks, with a particularly notable improvement of 1.3% in the object detection setting. We hope that these additional downstream experiments help address the reviewer’s concerns regarding the generality of the proposed approach.
>
> [1] Zheng, Sixiao, et al. "Rethinking semantic segmentation from a sequence-to-sequence perspective with transformers." Proceedings of the IEEE/CVF conference on computer vision and pattern recognition. 2021.
>
> [2] Zhang, Hao, et al. "Dino: Detr with improved denoising anchor boxes for end-to-end object detection." International Conference on Learning Representations, 2023.

---

### Official Review · Reviewer_B938 · 2025-10-28

**Soundness:** 3
**Presentation:** 3
**Contribution:** 2
**Rating:** 4
**Confidence:** 4

**Summary:**

This paper proposes UniSVD, a novel low-rank decomposition method for efficient compression of Transformer-based models. UniSVD exploits the linear nature of Multi-Head Attention by selectively decomposing only one side of each Q–K or V–O weight pair, while dynamically determining which side to factorize based on Frobenius-norm-based rank sensitivity analysis at the head and layer levels. This approach preserves information in rank-sensitive weights while significantly reducing parameter count and computational cost. Experiments on DeiT-Small and DeiT-Base models with the ImageNet-1K dataset demonstrate that UniSVD consistently improves accuracy and reduces latency compared to prior SVD-based methods such as FWSVD, ASVD, SVD-LLM, and FLAR-SVD, achieving over 40% parameter reduction with only a 2–3% accuracy drop. These results validate UniSVD as a simple yet generalizable low-rank decomposition framework that effectively balances efficiency and performance for Transformer architectures.

**Strengths:**

1. The proposed UniSVD maintains a level of parameter and computational efficiency comparable to existing combined decomposition methods, while introducing a unilateral decomposition strategy and a dynamically sensitivity-aware weight selection mechanism. Through this design, the method effectively preserves model accuracy without requiring any fine-tuning process.

2. The proposed UniSVD demonstrates strong practicality and applicability in real deployment and model compression scenarios by maintaining stable performance without any fine-tuning process. This highlights the proposed method’s contribution not only as a theoretically efficient approach but also as a training-free, low-cost compression framework that can be readily applied in real-world applications.


3. The theoretical explanation of the proposed method is presented in a logical and convincing manner. The mathematical formulation and analysis are well aligned with the underlying intuition, effectively supporting the validity and reliability of the proposed approach.

**Weaknesses:**

1. While Table 1 reports reductions in latency and GFLOPs, the paper does not provide sufficient analysis of the underlying causes. A more detailed explanation is needed to clarify which computational characteristics of the proposed method lead to reduced operations and latency. In addition, the paper lacks discussion on whether such acceleration effects are specific to certain hardware environments or can be consistently observed across different device types.


2. The proposed method leverages the characteristics of the MHA structure. Accordingly, it would be important to discuss whether this method can be extended to large-scale generative models such as Vision-Language Models (VLMs) or Large Language Models (LLMs) that also rely on MHA. Furthermore, the experiments are confined to the relatively simple DeiT–ImageNet setting, which limits the evaluation of the method’s generality and effectiveness for large-scale models. In particular, restricting the experiments to DeiT models does not sufficiently demonstrate the proposed method’s generalization capability, representing a critical limitation that diminishes the overall contribution of the work. As recent research trends emphasize validating applicability and generalization across diverse model architectures and datasets, broader experiments and validation are necessary to reinforce the paper’s claims.


Overall, the proposed method presents a novel and technically interesting idea that deserves recognition. However, empirical validation regarding the range of applicable models and the generalization performance remains limited. As mentioned in the future work, extending the approach to large-scale models such as LLMs or VLMs and demonstrating consistent generalization would significantly strengthen the contribution of this study. Incorporating additional experiments or analyses addressing these limitations in a future revision would greatly enhance the completeness and overall impact of the paper.

**Questions:**

1) The paper reports in Figure 4 that $W^{K}$and $W^{O}$are frequently selected due to their low rank sensitivity, and Table 3 shows that a fixed-selection strategy using these weights achieves nearly identical accuracy to the dynamic UniSVD variant. Given this, could the authors clarify whether the additional overhead of performing per-head SVD and loss computation twice in the dynamic selection process is practically justified, considering the marginal accuracy improvement observed?

---

> ### Author Response · Authors · 2025-11-21
> **Official Comment for Reviewer B938 (1/2)**
>
> ### **Details for Computational Benefits**
>
> We would like to clarify why the proposed method yields computational and parameter benefits over the conventional per-weight decomposition applied to attention layers. Existing approaches typically apply low-rank decomposition independently to each projection matrix in the attention block, *i.e*., to the Q, K, V, and O weights. In practice, this means that each original weight matrix is replaced by two low-rank sub-layers. As a result, (i) the number of linear layers per projection is doubled, and (ii) there is no reduction in parameters or computation unless the rank is reduced below 50% of the original dimension.
>
> For example, consider a square weight matrix of size $C×C$, and suppose it is decomposed with rank $r=C/2$ (*i.e*., 50% rank). The conventional approach factorizes this matrix into two tensors of size $C×r$ and $r×C$. The total number of parameters becomes $Cr + rC=2Cr=C^2$, which is identical to the original. Similarly, given an input $X∈R^{(N×C)}$, the original computation cost is $NC^2$. After decomposition, the cost is $NCr+NrC=2NCr=NC^2$, again yielding no computational savings.
>
> In contrast, the proposed method decomposes only one side and performs the tensor multiplications in advance. Suppose $K$ is selected and we again use $r=C/2$. Let the original projections be $W_q∈R^{(C×C)}$and $W_k=W_a W_b$, where $W_a∈R^{(C×r)}$and $W_b∈R^{(r×C)}$. By pre-multiplying, we obtain an equivalent formulation in which $W_q$ is replaced by $W_q W_a∈R^{(C×r)}$, and $W_k$ is reparameterized as a single matrix $W_c∈R^{(C×r)}$. In this setting, each of the Q and K projections is represented by a single $C×r$ matrix, so the parameters per weight are reduced from $C^2$ to $Cr$, corresponding to a 50% reduction when $r=C/2$. The associated computation similarly decreases from $NC^2$ to $NCr$, again yielding a 50% reduction. Although this example focuses on a specific case for clarity, the same mechanism underlies the parameter, FLOPs, and latency gains observed for the proposed method in our experiments.
>
> We hope that this explanation clarifies the source of the computational benefits and helps address the reviewer’s concerns.
>
> ---
>
> ### **Applicability and Generalization**
> - **Model : EVA**
> Method | Params (B) | GFLOPs|Top-1|
> -----|-------|-------|----|
> Baseline | 1.0 |620.6|	89.6 |
> Combined Weight Decomposition | 0.9 |550.5 |86.5|
> Unilateral Weight Decomposition |0.9 |550.5 |	**87.0** |
>
> - **Model : LLaVA1.5-13B**
> Method | Params (M) | GFLOPs|rand|pop|adv|
> -----|-------|-------|----|----|----|
> Baseline |303.5|191.1|87.4|87.2|84.2|
> Combined Weight Decomposition | 272.0 |167.8 |84.0	|82.7|80.1|
> Unilateral Weight Decomposition |272.0 |167.8|**84.1**|**82.9**|**80.5**|
>
> ***Table I. Comparison with the combined weight decomposition method and our UniSVD on vision-language models. Params for LLaVA1.5-13B indicates the parameters of the vision tower in LLaVA1.5-13B.***
>
> In addition to the experiments on diverse vision backbones described in the common response, we conducted further evaluations on VLM models (EVA and LLaVA1.5-13B) in response to Reviewer B938’s constructive comment. To specifically assess generality, we applied the proposed method only to the attention layers, without modifying the MLP blocks. We believe that combining our approach with existing MLP weight decomposition methods could further improve performance preservation in the future. EVA was evaluated on ImageNet, and LLaVA1.5 was evaluated on POPE benchmark, where our UniSVD was applied only to the vision encoder. For all methods, FLOPs were measured at an input resolution of (3, 336, 336).
>
> In Table I, the results indicate that our method can be applied to widely used VLM models. In particular, our method achieves around a 10% reduction in both parameters and FLOPs, while being more effective than the combined weight decomposition method in terms of performance. We hope that these additional VLM experiments help the reviewer better assess the generality of the proposed UniSVD approach.

---

> > ### Comment · Reviewer_B938 · 2025-11-24
> >
> > The proposed method shows promising results when applied to VLMs, which is indeed impressive.
> >
> > However, the first question that arises concerns the baseline parameter count reported for LLaVA1.5-13B.
> >
> > Table I states that the baseline model has 303.5M parameters, which suggests that the experiments may have been conducted solely on the visual encoder.
> >
> > If this is the case, I would appreciate clarification on why the proposed method was not applied to the LLM component as well, given that it constitutes the majority of the model’s parameters and computational cost.
> >
> > Furthermore, evaluating the method only on the POPE dataset appears insufficient to convincingly demonstrate its generalization capability.
> >
> > To more comprehensively support the claim of broad generalization, it would be beneficial to include evaluations on diverse task-oriented benchmarks such as VQA datasets (e.g., VQAv2, GQA), image–text retrieval datasets (e.g., Flickr30k, COCO), or multi-benchmark suites (e.g., MMBench, MME).
> >
> > Such additional experiments would help verify whether the method maintains stable performance across varying input distributions and more complex reasoning settings, thereby providing stronger evidence of its robustness and broad applicability.
> >
> > Lastly, I would like to note that one of my earlier questions remains insufficiently addressed.
> >
> > Figure 4 shows that certain projection weights are frequently selected due to their low rank sensitivity, and Table 3 indicates that a fixed-selection strategy using these weights achieves nearly identical accuracy to the dynamic UniSVD variant.
> >
> > Given this observation, I would appreciate clarification on whether the additional overhead of performing per-head SVD and computing the proxy loss twice in the dynamic selection process is practically justified, especially considering the marginal accuracy improvement it provides.

---

> > ### Comment · Reviewer_B938 · 2025-11-24
> >
> > The concern regarding the justification of the loss-computation overhead has been checked in the common response, and I find that it has been sufficiently addressed.

---

> ### Author Response · Authors · 2025-11-21
> **Official Comment for Reviewer B938 (2/2)**
>
> ### **Latency on Different Hardware Types**
> - **H100**
> Method | Params (M) | GFLOPs|Top-1 | Latency (ms)|
> -----|-------|-------|----|----|
> Baseline| 86.6|33.7|81.8|8.62|
> FLAR-SVD | 49.2|19.0|78.9|6.29|
> FLAR-SVD + UniSVD (Ours) |49.4|19.1|**79.3**|**5.96**|
>
> - **RTX 3090**
> Method|Params (M) | GFLOPs|Top-1 | Latency (ms)|
> -----|-------|-------|----|----|
> Baseline| 86.6|33.7|81.8|30.7|
> FLAR-SVD | 49.2|19.0|78.9|26.5|
> FLAR-SVD + UniSVD (Ours) |49.4|19.1|**79.3**|**23.5**|
>
> ***Table J. Performance and computations on the different types of hardware.***
>
> In response to the reviewer’s thoughtful comments, we measured latency on different types of GPU hardware (H100 and RTX 3090). As shown in Table J, our method (“FLAR-SVD + Ours”) consistently reduces latency compared to FLAR-SVD on both devices; for example, latency decreases from 6.29s to 5.96s on H100 and from 26.5s to 23.5s on RTX 3090, while also achieving better Top-1 accuracy. We hope that these results help address the reviewer’s concerns regarding hardware specific behavior.

---

> ### Author Response · Authors · 2025-11-26
> **Official Comment by Authors**
>
> ### **Additional Dataset Evaluations to Support Broad Generalization**
>
> | Method   | Params (M) | GFLOPs | VQAv2 |  GQA | TextVQA | MMBench |  MME   |
> |:--------:|:----------:|:------:|:-----:|:----:|:-------:|:-------:|:------:|
> | Baseline    |   303.5    | 191.1  | 78.5  | 61.9 |  58.2   |  64.6   | 1504.6 |
> | Combined weight decomposition|   272.0    | 167.8  | 73.6  | 59.4 |  53.1   |  63.0   | 1374.7 |
> | UniSVD (Ours)     |   272.0    | 167.8  | **74.7**  | **59.8** |  **53.1**   |  **63.0**   | **1408.0** |
>
> ***Table O. Comparison with the combined weight decomposition method and our UniSVD on various benchmarks. The reported parameter counts refer to the parameters of the vision encoder in LLaVA-1.5-7B.***
>
>
> | Method    | Params (M) | GFLOPs | R@1 (t→i) | R@5 (t→i) | R@10 (t→i) | R@1 (i→t) | R@5 (i→t) | R@10 (i→t) |
> |:---------:|:----------:|:------:|:---------:|:---------:|:----------:|:---------:|:---------:|:----------:|
> | Baseline      |   1136.6   | 299.9  |   74.6    |   92.3    |    95.2    |   90.0    |   98.6    |    99.4    |
> | Combined weight decomposition|   1021.2   | 267.6  |   73.7    |   91.3    |    95.1    |   88.3    |   98.3    |    99.4    |
> | UniSVD (Ours) |   1021.2   | 267.6  |   **74.0**    |   **91.5**    |    **95.1**    |   **89.3**    |   **98.3**    |    **99.4**    |
>
> ***Table P. Comparison with the combined weight decomposition method and our UniSVD on image–text retrieval dataset (Flickr30k). The reported parameter counts refer to the total parameters of EVA-CLIP. “t→i” denotes text-to-image retrieval, and “i→t” denotes image-to-text retrieval. R@K evaluates the percentage of queries for which the correct item is ranked within the top-K results.***
>
> We appreciate the reviewer’s suggestion to further validate the broad generalization of our method across additional datasets. In response, as shown in Table O, we conducted experiments on several VQA benchmarks (VQAv2, GQA, TextVQA) as well as multi-benchmark suites (MMBench, MME). The results show that our method reduces parameters and GFLOPs by 10.4% and 12.2%, respectively, while achieving better performance than the combined method We also observed that the degree of performance preservation varies across datasets. In particular, UniSVD exhibits notable improvements on VQAv2 and MME, whereas on MMBench the performance difference remains minimal.
>
> We additionally evaluate our method on the image–text retrieval benchmark, Flickr30k, using the EVA-CLIP model. As shown in Table P, our approach reduces parameters and FLOPs by 10.2% and 10.8%, respectively, while preserving performance significantly better than the combined method. Notably, in the image → text R@1 setting, our method achieves a +1.0% improvement over the combined baseline. For R@5 and R@10, the performance of UniSVD remains nearly identical to the combined strategy.
>
> We hope that these results adequately address the reviewer’s concerns, and we sincerely appreciate the suggestion to include additional validation experiments.

---

> > ### Comment · Reviewer_B938 · 2025-11-28
> >
> > While the proposed method demonstrates a modest reduction in parameters within the vision encoder, this compression appears to be limited due to the need to preserve accuracy. However, because the vision encoder accounts for only a small portion of the overall VLM workload, it is unclear how much this localized compression contributes to end-to-end inference acceleration. To properly assess the practical impact of the method, it would be helpful for the authors to clarify the extent to which the reduced vision encoder actually improves full-model inference speed or latency in real VLM scenarios.

---

> ### Author Response · Authors · 2025-11-26
> **Official Comment by Authors**
>
> ### **Clarification on Applying UniSVD**
>
> We acknowledge the reviewer’s careful observation that our method is applied only to the vision encoder, which is correct. We would also like to provide further clarification regarding the generality of the method and its current limitations in extending it to LLM components.
>
> Our approach is grounded in the linear operation between the Q–K and V–O projections in the attention module. Therefore, it is in principle applicable to any architecture that follows this attention mechanism. Consequently, its applicability extends beyond the DeiT setting and includes a wide range of vision encoders such as DeiT, ViT, and Swin, as well as VLMs that incorporate vision encoders (e.g., EVA, LLaVA). In addition, the method can be broadly used in downstream tasks such as semantic segmentation and object detection.
>
> Regarding attention mechanisms that incorporate RoPE[1], we note that the rotary matrix introduced by RoPE can still be expressed as a linear transformation, as shown in the following formulation:
>
> $q^T_mk_n = (R^d_{\Theta,m}W_qx_m)^T(R^d_{\Theta,n}W_kx_n)=x^TW_qR^d_{\Theta,n-m}W_kx_n \ ,$
> where $R^d_{\Theta,m}$ is the rotary matrix with $\Theta=$ {$\theta=10000^{-2(i-1)/d}, i \in [1,2,...,d/2]$} and $R^d_{\Theta,n-m}=(R^d_{\Theta,m})^TR^d_{\Theta,n}$. Note that $R^d_{\Theta}$ is an orthogonal matrix. $q_m$ indicates the query that incorporates the $m^{th}$ position and $k_n$ indicates the key that incorporates the $n^{th}$ position.
> This preserves the applicability of our decomposition strategy within RoPE-based attention.
>
> However, in the case of LLMs, the model operates with a recurrent mechanism in which the sequence length changes dynamically during inference. As a result, the value of the rotary matrix $R^d_{\Theta,n-m}$ also changes at every input step. Since this behavior makes it challenging to anticipate or prepare for all sequence lengths in advance, applying our method to LLMs becomes difficult in practice.
>
> Nevertheless, when considering the possibility of future extensions, we note that recent work [2] has explored replacing layer normalization in GPT-2 with a linear transformation during inference. This study was submitted to ICLR 2026 and received very positive evaluations. If such approaches become more common in LLMs, increasing the number of stages where linear operations between weights are preserved, we expect that the applicability of our method to LLM architectures may expand in the future. We hope that our additional response addresses the reviewer’s concerns.
>
>
> [1] Su, Jianlin, et al. "Roformer: Enhanced transformer with rotary position embedding." Neurocomputing 568 (2024): 127063.
>
> [2] Baroni, Luca, et al. "Transformers Don't Need LayerNorm at Inference Time: Scaling LayerNorm Removal to GPT-2 XL and the Implications for Mechanistic Interpretability." arXiv preprint arXiv:2507.02559 (2025).

---

> ### Author Response · Authors · 2025-11-28
> **Official Comment by Authors**
>
> ### **Additional Latency Evaluation and UniSVD Effects on VLMs**
>
> Method|Params (M)|GFLOPs|Latency (ms)
> ----|----|----|----|
> EVA-CLIP (Baseline) | 1136.6 | 299.9 |52.2
> |+ UniSVD (Ours)|1021.2|267.6|46.5
>
> ***Table R. Computations and latency of our UniSVD on EVA-CLIP.***
>
> Method|Vision Params (M)|Vision GFLOPs| Latency (ms)
> ----|----|----|----|
> LLaVA-1.5-7B (Baseline)|303.5	|191.1|215.2
> |+ UniSVD (Ours)|272.0|167.8|209.3
>
> ***Table S. Computations and latency of our UniSVD on LLaVA1.5.***
>
> We sincerely appreciate the reviewers thoughtful and constructive feedback. In response to these comments, we conducted additional latency evaluations for both EVA-CLIP and LLaVA-1.5-7B, as presented in Tables O and P. EVA-CLIP was evaluated on the Flickr30k benchmark, while LLaVA was assessed using the POPE dataset, both measured on an RTX 3090 GPU. The results show that EVA-CLIP, where the vision encoder accounts for a substantial portion of the computational workload, experiences a meaningful latency improvement of approximately 10.9%. In contrast, for LLaVA-1.5-7B, where the vision encoder contributes only a small fraction to the total computation, the improvement is measured at 2.8%, which is consistent with the architectural characteristics of the model.
>
> Although the main scope of our paper is vision models, we carried out additional experiments with vision language models in accordance with the reviewers suggestion to examine generality more thoroughly. For this purpose, we applied our weight decomposition method only to the attention modules within the vision encoder of the VLM. This design choice ensures a fair and minimally intrusive extension of our method to VLMs, compared to the combined method, while allowing us to meaningfully evaluate its generalization capability.
>
> Our method is designed as an additive component that can work together with existing weight decomposition approaches across all layers. Therefore, when combined with weight decompositions applied to the MLP layers, it is expected to yield greater reductions in both parameters and floating point operations. However, many of the existing techniques discussed in the paper, such as ASVD and SVD-LLM, rely heavily on calibration sets. While our draft follows the calibration strategy used in FLAR-SVD, the application of these baselines to VLMs would require the construction of new calibration sets, which is a non trivial and model dependent process. For this reason, and to focus specifically on generality, we applied only our method in the VLM experiments. We believe that this approach holds meaningful potential as future work for extending our approach to broader VLM tasks.
>
> We hope that these additional results provide valuable evidence that our approach can be extended to more complex tasks and larger multimodal architectures. We are grateful to the reviewer for motivating us to further explore this challenging field, which has significantly strengthened the contribution of our work.

---

### Official Review · Reviewer_1opc · 2025-10-30

**Soundness:** 3
**Presentation:** 4
**Contribution:** 2
**Rating:** 2
**Confidence:** 4

**Summary:**

This paper proposes UniSVD, a simple and training-free low-rank compression method for attention-based vision transformers. Instead of decomposing all attention weights, UniSVD decomposes only one side of each Q/K or V/O pair, based on which side is less sensitive to rank reduction. This “unilateral” strategy keeps more information from the important weights and reduces FLOPs and parameters. Experiments on DeiT-Small and DeiT-Base show consistent improvements over several existing SVD-based decompositions, without any fine-tuning.

**Strengths:**

UniSVD offers a novel insight into attention decomposition: discovering and utilizing the asymmetric sensitivity of Q/K and V/O matrices to improve compression without retraining.
The method is conceptually simple and easy to follow, requiring only SVD and a dynamic selection rule based on Frobenius loss.

**Weaknesses:**

The experimental results mainly highlight two aspects:
(1) UniSVD is orthogonal to existing SVD-based decompositions and consistently improves upon them;
(2) it outperforms simpler variants such as Per-Weight Decomposition and Combined Weight Decomposition.
However, the evaluation lacks comparison against stronger and more recent low-rank decomposition baselines from the literature, making it unclear whether UniSVD achieves SOTA performance beyond the specific SVD variants tested here.

From a methodological perspective, while the core intuition is reasonable, the technical novelty is modest. The method essentially adds a dynamic selection between Q/K and V/O before applying SVD. The results suggest that decomposing K over Q and O over V tends to be optimal, and the additional head or layerwise dynamic selection provides only marginal benefit, since Table 3 shows negligible differences.

|        | 20%  | 40%  | 60%  | 20%  | 40%  | 60%  | 20%  | 40%  | 60%  |
|--------|-------|-------|-------|-------|-------|-------|-------|-------|-------|
| K-O  | 78.3  | 74.2  | 65.3  | 79.6  | 76.1  | 69.3  | 81.7  | 80.3  | 71.7  |
| DynamicSelection | 78.4  | 74.4  | 65.4  | 79.9  | 76.5  | 69.3  | 81.7  | 80.4  | 72.1  |

Thus, the main contribution appears to lie in identifying the relative sensitivity of Q/K/V/O weights to decomposition rather than introducing a fundamentally new decomposition framework.

Finally, the method is evaluated only on DeiT models and ImageNet classification. It remains uncertain whether UniSVD generalizes to other ViT architectures e.g. Swin Transformer, downstream tasks such as detection or segmentation, or also transformer-based LLMs. If demonstrated, such generalization could substantially strengthen the paper’s contribution and broader impact.

**Questions:**

q1: In your experiment, why do you use the distilled version of DeiT-S only but not the other two model sizes?
q2: In figure 4, it seems that in most cases, one component will dominate the other choice. so I'm curious how the following settings works: just calculate per-layer error and always choose the dominant choice regardless of the head.
q3: I'm curious about what is the ratio of choosing Q/K and choosing V/O, it seems that K is dominating Q and O is dominating V.
q4: What is the computational cost of the head-wise dynamic selection during inference

---

> ### Author Response · Authors · 2025-11-21
> **Official Comment for Reviewer 1opc (1/2)**
>
> Thanks for the reviewer 1opc's constructive comments. We addressed concerns about 'Generalize to other architectures' in **[Common Response 1 & 2]**, and concerns about 'Computational cost of the head-wise dynamic selection during inference' in the **[Common Response 3]**.
>
> ---
>
> ### **Generality Across Diverse Downstream Tasks**
>
> - **Semantic Segmentation**
> Method | Params (M) | GFLOPs|mIoU (%)|
> ---|---|---|---|
> Baseline |305.5|959.6|77.4|
> Combined Weight Decomposition |280.3 |836.3 |76.8|
> Unilateral Weight Decomposition |280.3 |836.3 |**76.9**|
>
> ***Table E. Weight decomposition comparison on Cityscapes***
>
> - **Object Detection**
> Method | Params (M) | GFLOPs|mAP|
> ---|---|---|---|
> Baseline | 195.2|756.9|57.2|
> Combined Weight Decomposition | 184.6|680.4|54.1 |
> Unilateral Weight Decomposition |184.6 |680.4 |**55.4**|
>
> ***Table F. Weight decomposition comparison on COCO***
>
> In addition to the experiments on other backbones described in the common response, we also conducted experiments on downstream tasks in Table E and F, including semantic segmentation and object detection. For semantic segmentation, we applied the proposed method to SETR [1], which uses ViT as its backbone. For object detection, we used DINO [2] with a Swin Transformer backbone. To specifically assess generality, we applied our method only to the attention phases. The results indicate that our method is more effective than the combined weight decomposition method on both tasks, with a particularly improvement of 1.3% in the object detection setting. We hope that these additional downstream experiments help address the reviewer’s concerns regarding the generality of the proposed approach.
>
> [1] Zheng, Sixiao, et al. "Rethinking semantic segmentation from a sequence-to-sequence perspective with transformers." Proceedings of the IEEE/CVF conference on computer vision and pattern recognition. 2021.
>
> [2] Zhang, Hao, et al. "Dino: Detr with improved denoising anchor boxes for end-to-end object detection." International Conference on Learning Representations, 2023.
>
> ---
>
> ### **Discussion of the Limited Comparison with SOTA Methods**
>
> We agree with the reviewer that comparisons with more recent SOTA methods are limited. At the same time, the main goal of this work is to propose an add-on module that can be applied to the attention layers on top of existing strong decomposition methods. Accordingly, we evaluated the proposed add-on in combination with FLAR-SVD, which was introduced for vision models, and ASVD/SVD-LLM, which were proposed in the LLM domain. Since weight decomposition has been actively studied in LLMs and similar per-weight decomposition frameworks are now used in the attention layers of vision models, this work is motivated by the need to replace that attention-side decomposition with a more efficient add-on.
>
> From a strict SOTA perspective, to the best of our knowledge, FLAR-SVD currently achieves the strongest results among training-free decomposition methods for vision models. In the LLM domain, ModeGPT and SVD-LLM v2 are among the most competitive approaches. However, ModeGPT is dependent on specific LLM architectures (e.g., gated MLP structures), which makes a naïve and faithful adaptation to vision backbones difficult. SVD-LLM v2, on the other hand, does not yet provide publicly available code, so an accurate implementation and comparison would be unreliable. Within these constraints, we focused on methods that can be reasonably and reproducibly applied. If there are additional comparisons that the reviewer believes are practically feasible, we would be very willing to conduct those experiments.

---

> ### Author Response · Authors · 2025-11-21
> **Official Comment for Reviewer 1opc (2/2)**
>
> ### **Technical Novelty Related to Fixed-Selection Ablation**
> | Method | Small 40% | Small 60% | Small distill 40% | Small distill 60% | Base 40% | Base 60% |
> |---|:-----:|:-----:|:---:|:----:|:----:|:----:|
> | Per-weight Decomposition | 67.2  | 31.4  | 56.2 | 23.0 | 79.0  | 71.4  |
> | Combined Weight Decomposition | 74.2  | 54.2 | 76.1  | 63.6  | 79.4 | 65.7  |
> | UniSVD – KO Ablation  | 74.2  | 65.3  | 76.1  | 69.3  | 80.3  | 71.7  |
> | UniSVD – Dynamic  | 74.4  | 65.4  | 76.5  | 69.3  | 80.4   | 72.1 |
>
> ***Table G. Comparison with conventional weight decomposition with our UniSVD framework***
>
> We thank Reviewer 1opc for raising this point. Your comment led us to carefully revisit the implications of the results in Table 3 of our draft, which we also consider important. The key point we aim to convey here is that the combined decomposition method is not optimal. Motivated by this, we propose the UniSVD framework, which decomposes only one side (either QK or VO). In this sense, the KO fixed selection can be regarded as a special case within the UniSVD framework. As shown in Table G, even the KO fixed selection already achieves better performance than our baseline combined method.
>
> However, when extending the proposed method to different tasks or models, we cannot guarantee what tensor properties the pretrained QK and VO weights will exhibit. For this reason, the dynamic selection is designed to always achieve the minimum approximation error under our objective, even if the improvement over the KO fixed selection is sometimes relatively small. We would greatly appreciate it if the reviewer could view the dynamic method as a way to ensure that our method behaves adaptively and reliably across diverse architectures, rather than as a mechanism tuned only for the specific setting reported in Table 3.
>
> ---
>
> ### **Selection Ratio of Q-K and V-O**
> | Model    | Q    | K     | V   | O  |
> |----|-----|------|-------|-----|
> | DINOv2 | 22 (15%)   | 122 (85%)  | 16 (11%) | 128 (88%)  |
> | ViT-Large  | 115 (30%) | 269 (70%)  | 28 (7%)| 356 (93%)  |
> | Swin-Large | 197 (35%)  | 367 (65%)  | 192 (34%) | 372 (66%)   |
> | DeiT-Large | 67 (17%)  | 317 (83%)  | 116 (30%)  | 268 (70%)  |
>
> ***Table H. Selection ratio of Q-K and V-O on various models.***
>
> To address the reviewer’s question, we additionally analyzed the selection ratio of QK and VO for each model in Table H. Consistent with the reviewer’s observation, K and O are indeed selected more frequently overall. However, they are not overwhelmingly dominant, and the exact ratios vary across models.
>
> We also acknowledge that **Figure 4** of our draft may have caused confusion. When the approximation errors of the two choices are very close, the curves for KO can overlap and appear above those for QV in the plot, making some QV-selected cases look as if KO had been selected instead. We agree that this visualization can give the impression that KO are more dominant than they actually are. To reduce this misunderstanding, we will incorporate the table-style presentation of selection ratios suggested by the reviewer in the revised draft. We appreciate the constructive feedback.
>
> ---
>
> ### **Clarification of Head-wise vs. Layer-wise Methods and Additional Experiments with the Per-Layer Error method**
>
> In the current version of the paper, the description “head- and layer-wise” is inaccurate, and it should indeed be described as head-wise. Our intention was to emphasize that the Q-K and V-O selection can vary across heads at each layer, but this was not clearly reflected in the wording. We will correct this expression in the revised draft to accurately describe the method, reflecting the reviewer’s constructive feedback.
>
> Furthermore, following the reviewer’s suggestion, we conducted an additional layer-wise experiment, where the selection was performed by aggregating the approximation errors over all heads within each layer. In this setting, as shown in **Figure 12** of our draft, Q-V tended to be selected in the early layers, while K-O was consistently selected in the middle and later layers. In terms of performance, the original UniSVD (with head-wise selection) achieved slightly better results than this layer-wise variant. We appreciate this interesting suggestion and hope that these additional experiments help clarify the behavior of our method with respect to head-wise and layer-wise selection.

---

> ### Author Response · Authors · 2025-11-27
> **Official Comment for Reviewer 1opc**
>
> ### **Additional Experiment: Applying UniSVD to More Recent and Stronger Baseline**
>
> | Method              | Params | GFLOPs | Top-1 | Latency |
> |:-------------------:|:------:|:------:|:-----:|:-------:|
> | SVD-LLM             | 44.1   | 17.0   | 70.3  | 26.7    |
> | SVD-LLM v2          | 44.1   | 17.0   | 70.5  | 26.3    |
> | SVD-LLM v2 + UniSVD | 44.1   | 17.0   | **72.8**  | **25.2**    |
>
> ***Table Q. Comparison of SVD-LLM, SVD-LLM v2, and SVD-LLM v2 with our UniSVD on DeiT-Base for ImageNet-1K.***
>
> We appreciate the reviewer’s insightful comment regarding the limited comparison against stronger and more recent low-rank decomposition baselines. Although we addressed this point in the “Discussion of the Limited Comparison with SOTA Methods” section, we further conducted additional experiments to more thoroughly resolve the reviewer’s concern.
>
> Among training-free low-rank decomposition methods, FLAR-SVD[1] currently achieves state-of-the-art performance in the vision domain (already included in our main comparison), while SVD-LLM v2[2] was recently introduced as a stronger baseline for LLMs. Since the official implementation of SVD-LLM v2 has not yet been released, we re-implemented it on DeiT-Base for ImageNet-1K evaluation by strictly following the methodological description in the paper. For fairness, we kept the rank-search procedure identical to SVD-LLM v1 and replaced only the SVD truncation scheme with the loss-optimized weight truncation proposed in SVD-LLM v2.
>
> We compare three variants: SVD-LLM v1[3], SVD-LLM v2, and SVD-LLM v2 + UniSVD applied to the attention layers. The results show that SVD-LLM v2 yields only a marginal improvement of about 0.2 % over v1 in the vision setting. In contrast, applying our UniSVD to the SVD-LLM v2 pipeline results in a substantial accuracy gain of 2.5 percent and 2.3 percent over v1 and v2, respectively, while maintaining similar parameters, FLOPs, and latency. These results show that our proposed UniSVD can be effectively applied even to more recent and stronger baselines while still achieving solid performance improvements. We hope that this additional experiment helps address the reviewer’s concern. If there are any remaining issues that we may not have fully addressed, we would be more than happy to clarify them in detail within the rebuttal period.
>
> [1] Thoma, Moritz, et al. "FLAR-SVD: Fast and Latency-Aware Singular Value Decomposition for Model Compression." Proceedings of the Computer Vision and Pattern Recognition Conference. 2025.
>
> [2] Wang, Xin, et al. "Svd-llm v2: Optimizing singular value truncation for large language model compression. Conference of the Nations of the Americas Chapter of the Association for Computational Linguistics. 2025.
>
> [3] Wang, Xin, et al. "Svd-llm: Truncation-aware singular value decomposition for large language model compression." International Conference on Learning Representations. 2025.

---

### Official Review · Reviewer_3Es2 · 2025-11-01

**Soundness:** 2
**Presentation:** 3
**Contribution:** 2
**Rating:** 4
**Confidence:** 3

**Summary:**

This paper introduces UniSVD, a novel weight decomposition method for compressing attention modules in vision transformers. Unlike conventional approaches that decompose all weight matrices or combined weight products, UniSVD selectively decomposes only one side of the Q-K and V-O weight pairs based on their sensitivity to low-rank approximation. The selection is performed dynamically at the head and layer level using Frobenius norm-based approximation error. The authors demonstrate consistent improvements when UniSVD is integrated with existing SVD-based compression methods on DeiT models.

**Strengths:**

1 - The unilateral decomposition idea cleverly exploits the linear nature of Q-K and V-O operations in attention mechanisms.

2 - Table 1 shows substantial gains across multiple baseline methods (e.g., 8.5% improvement for ASVD on DeiT-Small).

3 -  The paper includes thorough ablation studies examining dynamic vs. fixed selection strategies and head-wise selection patterns.

**Weaknesses:**

1. **Limited scope and evaluation**:
   - Only evaluated on DeiT models with ImageNet-1K
   - No experiments on larger/recent models (ViT-Large, DINOv2, etc.)
   - Cannot be applied to MLP layers, limiting overall compression potential

2. **Lack of theoretical analysis**: No formal analysis of why unilateral decomposition preserves more information than alternatives. The singular value analysis in Figure 2 is superficial and doesn't rigorously support the claims.

3. **Computational overhead concerns**: The dynamic selection requires computing SVD for both weights to make decisions, potentially negating efficiency gains. This overhead is not analyzed.

4. **Missing critical comparisons**:
   - No wall-clock time measurements for the selection process
   - Limited baseline comparisons (only SVD-based methods)

5. **Inconsistent experimental details**:
   - Table 1 shows different parameter counts for FLAR-SVD (49.2M) vs others (44.1M) without explanation
   - The "hierarchical strategy" for rank assignment is mentioned but not detailed

6. **Weak empirical analysis**: The claim that combined weights have "noticeably larger singular values" (Figure 2) lacks quantitative support and statistical testing.

**Questions:**

1. What is the actual computational overhead of the dynamic selection process? Computing SVD for both weights seems expensive. Can you provide wall-clock time comparisons?

2. Why limit evaluation to DeiT models? Have authors tested on ViT-Large, Swin Transformers, or recent models like DINOv2? How does the method scale?

3. Can authors please provide a theoretical analysis?  Why should unilateral decomposition preserve more information than combined decomposition from an information-theoretic perspective?

4. How does UniSVD compare with structured pruning or quantization methods in terms of accuracy-efficiency trade-offs?

5. What is the sensitivity to rank selection? How robust is the dynamic selection to different target ranks?

6. Why do parameter counts differ in Table 1 between FLAR-SVD and other methods for the same model?

7. Can you provide quantitative analysis for Figure 2? Statistical tests comparing singular value distributions would strengthen your claims.

---

> ### Author Response · Authors · 2025-11-21
> **Official Comment for Reviewer 3Es2 (1/2)**
>
> Thanks for the reviewer 3Es2's insightful comments. We addressed concerns about 'Limited scope and evaluation' in **[Common Response 1 & 2]**, and concerns about 'Computational overhead' & 'Wall-clock time measurements for the selection process' in the **[Common Response 3]**.
>
> ---
>
> ### **Details on Parameter Count**
>
> Unlike methods (FWSVD, ASVD and SVD-LLM) that adjust the rank by the same ratio for each layer without using a rank search, FLAR-SVD uses the rank search technique they proposed to determine the rank for each layer. During their search process, if the search loss falls below the threshold, the rank for that layer is incrementally increased. Consequently, FLAR-SVD possesses different parameters compared to other methods.
>
> ---
>
> ### **Details on Hierarchical Strategy**
>
> We thank the reviewer for the comments on the hierarchical strategy and regret that our original description was not sufficiently clear. In this paper, the hierarchical strategy used for rank assignment is in fact very simple: given a predefined pair $(R_{min},R_{max})$, the ranks are assigned layer-wise according to an arithmetic progression between these two values. Since our method is applied to the attention modules of an existing approach, $(R_{min},R_{max})$ is chosen so that the overall parameter count is aligned with that of the corresponding baseline method. This means that no sophisticated rank-search technique is employed; the reported performance is achieved with a naïve rank assignment strategy. We believe there is clear potential for further improvement if our method is combined with more advanced rank selection schemes, which we consider a promising direction for future work. We again thank the reviewer for pointing out this ambiguity, and we have revised the draft to explicitly clarify the details of the hierarchical strategy.
>
> ---
> ### **Quantitative analysis of singular values with discussion of theoretical analysis**
> | Method   | Rank | MHA param reduction (%) | Top-1 Acc. |
> |------|------|-------|------|
> | Combined Weight Decomposition  | 16   | 75   | 19.1  |
> | Unilateral Weight Decomposition (Ours) | 16   | 75   | **41.9**  |
> | Combined Weight Decomposition   | 24   | 62.5  | 63.6   |
> | Unilateral Weight Decomposition (Ours)  | 24   | 62.5  | **69.5** |
>
> ***Table C. Performance under singular value analysis in Appendix Figure 6.***
>
> **Experiment of singular value.**
> We thank the reviewer for raising this point and for suggesting a new analysis direction. As suggested, we quantified the difference between the singular values of the combined matrix and the selected individual matrix, and reported the results in **Figure 6 of the revised draft**. Specifically, for ranks 16 and 24, we computed the cumulative singular values of the combined weight and of the selected individual weight for each layer, and then measured their differences in Figure 6. We observed that in some low-level layers the combined method can indeed have a smaller cumulative singular value. However, when we look at the performance in Table C, the proposed UniSVD still achieves a significant improvement over the combined method. We believe that this singular value–based analysis provides useful empirical insight for future work, and we appreciate the reviewer’s suggestion.
>
> **Discussion of theoretic analysis.**
> An additional point we would like to emphasize is that the singular values are compared in different spaces, namely on the combined matrix versus the individual matrices, which makes a fair comparison difficult. To obtain a more consistent view, we mapped the combined matrix into the individual space and analyzed the approximation error there, as shown in Figure 3 of the revised draft. In this unified space, the proposed method exhibits lower approximation error than the combined method, and this corresponds well to the observed performance improvements. Based on these results, our empirical conclusion is that optimizing in the individual space is more beneficial than optimizing in the combined space in this setting. At the same time, as the reviewer rightly pointed out, there is still room to further deepen our theoretical understanding of why optimization in the individual space is advantageous. We view the reviewer’s suggestion of singular value–based numerical analysis as a promising starting point for developing a deeper theoretical explanation, and believe this will be a fruitful direction for future work. If the reviewer has further ideas on how this analysis could be extended, we would be very grateful for additional guidance and would be happy to explore them.

---

> ### Author Response · Authors · 2025-11-21
> **Official Comment for Reviewer 3Es2 (2/2)**
>
> ### **Robustness of Dynamic Selection Across Different Ranks**
>
> To analyze how the dynamic selection behaves under different target ranks and to assess its robustness, we added an additional study in the revised draft (*i.e*., **left of Figure 14**). In this analysis, we compare the proposed dynamic scheme with several fixed-selection variants and report performance for a range of rank values. For each point on the x-axis, all heads are assigned the same rank, which allows us to directly examine the sensitivity of performance to the chosen rank.
>
> The results show that, as the rank decreases, the proposed dynamic method consistently maintains more robust performance than the fixed-selection baselines. We hope that this additional analysis helps clarify the behavior of the dynamic scheme and addresses the reviewer’s question.
>
> ---
>
> ### **Comparison with Other Compression Tasks: Quantization and Pruning**
> | Compression | Method  | Param (↓%) | GFLOPs (↓%) | W/A | Top-1 Acc. |
> |-----|-------|-----:|------:|:---:|-----:|
> |       | Base        |     –      |     –      |  –  |    81.8    |
> | Pruning w/o fine-tuning   | Ours   |   43.0     |   43.3     |  –  |    79.3    |
> |       | FLAR-SVD    |   43.2     |   43.6     |  –  |    78.9    |
> | Pruning w/ fine-tuning    | CT-GFM      |   40       |     –    |  –  |   81.28   |
> |     | MD-ViT      |     –      |   60       |  –  |    81.5    |
> |    | UVC         |     –      |   54.5     |  –  |    80.57   |
> | Token pruning (w/ fine-tuning) | DynamicViT |     –      |   36   |  –  |  81.3    |
> |       | EVO-ViT     |     –      |   33       |  –  |    81.3    |
> | Quantization  | PTQ4ViT     |     –      |     –      | 4/4 |    64.39 |
> |       | APQ-ViT     |     –      |     –      | 4/4 |    67.48   |
> |     | RepQ-ViT    |     –      |     –      | 4/4 |    75.61   |
> |     | ERQ         |     –      |     –      | 4/4 |    78.23   |
> |    | IGQ-ViT     |     –      |     –      | 4/4 |    79.23   |
> |       | AdaLog      |     –      |     –      | 4/4 |    78.03   |
> |     | I&S-ViT     |     –      |     –      | 4/4 |    79.97   |
> |     | DopQ-ViT    |     –      |     –      | 4/4 |    80.13   |
> |        | QDrop*      |     –      |     –      | 4/4 |    79.96   |
> |        | OASQ        |     –      |     –      | 4/4 |    78.83   |
> |    | FIMA-Q      |     –      |     –      | 4/4 |    80.33   |
>
> ***Table D. Comparison with various pruning and quantization methods.***
>
> To address the reviewer’s question, we provide a comparative table including various pruning and quantization approaches in Table D. Existing pruning methods generally achieve strong performance, but most of them rely on fine-tuning after pruning. Recent quantization methods with 4-bit weights and activations have started to reach performance close to the original (full-precision) models. In contrast, the proposed decomposition-based, training-free weight pruning approach has only recently been explored in the vision domain, and this line of work is still relatively underdeveloped. We hope that this comparison with other compression techniques helps clarify the position and characteristics of our method and addresses the reviewer’s concerns.

---

### Author Response · Authors · 2025-11-21
**Common Response for Reviewers 3Es2, 1opc, B938, Rygs (3/3)**

### **[Common Response.3] Additional Overhead of Dynamic Selection Compared to Fixed Selection**

| Model  | DINOv2| ViT-Large | Swin-Large| DeiT-Large|
|------ |--------:|----:|----:|----:|
| w/o selection (sec) | 0.6 |  1.3 |  1.1 | 1.3 |
| w/ selection (sec) | 1.2 | 2.6 | 2.4 | 2.6 |

***Table B. Wall-clock time comparison for the dynamic selection. Each time is the cumulative time for all layers. Results are measured on a single 3090 GPU.***

First, we would like to clarify that **the proposed dynamic selection method does not introduce additional overhead at inference time compared to the fixed selection method**. Because the selection is defined on the decomposed weights, all selected components can be precomputed and initialized once, and then reused for multiple inference runs without extra per-sample computation. Therefore, in our dynamic method, the selection is not adaptively recomputed for each input sample. Instead, our method operates dynamically over the per-head weights of each model.

To further alleviate the reviewers’ concern about the overhead of this dynamic process, we measured the processing time in a setting where the pre-computation step is explicitly included, as reported in Table B. This measurement covers the entire procedure, including weight decomposition and the selection required to obtain the precomputed weights. The results show that the dynamic selection method incurs *only a minor overhead of up to 0.6~1.3 seconds* compared to the fixed selection baseline, which we believe sufficiently addresses the concern about the additional overhead.

---

### Author Response · Authors · 2025-11-21
**Common Response for Reviewers 3Es2, 1opc, B938, Rygs (2/3)**

### **[Common Response.2] Discussion of UniSVD Limitations and Extensibility**

We appreciate the reviewer’s concerns regarding the generality of the proposed method. To address this point, we added experiments on diverse architectures and tasks in **Table 2,4,5 and Section 4.7 of the revised draft**, including **additional vision backbones, VLMs, semantic segmentation, object detection and semantic segmentation**. Beyond these empirical results, we provide a more detailed discussion of the applicability of our approach below. The proposed method is built on the linearity between the Q–K and V–O projections in the attention module. Therefore, it is in principle applicable to any architecture that follows this attention mechanism.

First, this includes *widely used vision encoders*, such as Vision Transformers, DeiT, Swin, and PVT, where the attention blocks are implemented with linear Q, K, V, and O projections.

Second, the method can be *applied to VLMs* such as EVA and LLaVA-1.5, which employ these vision encoders as their visual backbone.

Third, the method is also *applicable to ROPE-based attention* when the input sequence length is fixed. For example, in video generation models such as Hunyuan [1], the input sequence length is fixed, so the RoPE transformation applied to Q and K can be absorbed into a fixed linear transformation. In this setting, one can treat the rotation matrices as constant and perform weight decomposition accordingly.

At the same time, we acknowledge an important limitation. For LLMs using RoPE with variable input lengths, it is not feasible to precompute optimal decompositions for all possible rotary matrices, which makes direct application of our method difficult. We believe that clearly stating this applicability condition and limitation better reflects the true generality of the proposed approach.

However, even for LLMs, there are promising directions for extending our method. A very recent work [2], submitted to ICLR 2026, reports that layer normalization is not strictly required at inference time for GPT-2–style models, and this submission received very strong preliminary scores. This study replaces all layer normalization layers in the QKV projections and MLP blocks with equivalent linear transformations. Under such architectures, many additional linear relationships between adjacent weights naturally arise, and our method can be applied to these linear layers as well. Therefore, we believe that the proposed approach has substantial potential to be further extended and applied to LLMs as these architectures become more widespread.

Overall, we believe that the additional experiments and the above clarifications demonstrate that **the proposed method is broadly applicable to a wide range of attention-based vision and vision-language models**, while also honestly acknowledging its current limitations for LLM settings. If the perceived generality of the method was a major factor in your evaluation, we hope that this response helps alleviate your concerns, and we would be grateful if you could take these points into consideration when updating your score.

[1] Kong, Weijie, et al. "Hunyuanvideo: A systematic framework for large video generative models." arXiv preprint arXiv:2412.03603 (2024).

[2] Baroni, Luca, et al. "Transformers Don't Need LayerNorm at Inference Time: Scaling LayerNorm Removal to GPT-2 XL and the Implications for Mechanistic Interpretability." arXiv preprint arXiv:2507.02559 (2025).

---

### Author Response · Authors · 2025-11-21
**Common Response for Reviewers 3Es2, 1opc, B938, Rygs (1/3)**

We sincerely appreciate all reviewers for their careful reading of our paper and for the constructive comments and suggestions. We have carefully addressed all concerns raised in the reviews and updated our draft where appropriate to reflect the corresponding clarifications and additional experiments. New materials (**Tables 2,4-5,7-9 and Figures 6,12-14**) are highlighted in blue to aid reviewers. We hope that our responses sufficiently resolve the raised issues, and if there are any further points that require clarification, we would be willing to address them in detail during the rebuttal period.

---

### **[Common Response.1] Extensibility of UniSVD on Recent and Larger Models**
- **DINOv2**
| Method | MHA Params (M) | Top-1 | Latency (ms) |
|-----|------:|------:|------:|
| Baseline  | 28.3  | 81.2  | 38.6 |
| Per-Weight Decomposition| 22.5   | 65.5  | 37.3 |
| Combined Weight Decomposition | 22.5  | 64.2  | 36.0 |
| UniSVD  | 22.5  | **73.2**  | 35.7 |

- **ViT-Large**
| Method | MHA Params (M) | Top-1 | Latency (ms) |
|-----------|-------:|------:|--------:|
| Baseline  | 100.7 | 84.3  | 100.6 |
| Per-Weight Decomposition | 50.3 | 75.0  | 91.4 |
| Combined Weight Decomposition | 50.3  | 74.9  | 83.6 |
| UniSVD | 50.3  | **77.7** | 81.1 |

- **Swin-Large**
| Method | MHA Params (M) | Top-1 | Latency (ms) |
|-----------|-----------:|------:|--------:|
| Baseline  | 62.8  | 86.3  | 67.8    |
| Per-Weight Decomposition | 31.4  | 80.4  | 63.9    |
| Combined Weight Decomposition | 31.4 | 79.2  | 58.3    |
| UniSVD    | 31.4       | **81.2**  | 57.3    |

- **DeiT-Large**
| Method | MHA Params (M) | Top-1 | Latency (ms) |
|--------|-----:|------:|--------:|
| Baseline  | 100.7 | 86.8  | 99.1  |
| Per-Weight Decomposition | 50.3  | 82.3  | 91.7    |
| Combined Weight Decomposition | 50.3  | 81.7  | 84.5    |
| UniSVD    | 50.3 | **83.0**  | 81.9    |

***Table A. Comparison with the conventional weight decomposition methods on various vision models.***

To address the constructive comments from Reviewers 3Es2, 1opc, B938, and Rygs, we conducted additional experiments on larger and more recent vision models in Table A. Specifically, we applied UniSVD to DINOv2, ViT-Large, Swin-Large, and DeiT-Large. The results show that our method can be effectively applied to these diverse architectures, while preserving performance more effectively than both per-weight and combined weight decomposition strategies. We believe these experiments help to better assess the scalability and extensibility of UniSVD. A more detailed discussion of the scalability of UniSVD is provided in ‘Discussion of UniSVD Limitations and Extensibility’ Section of our rebuttal.

---

### Author Response · Authors · 2025-12-02
**Final Clarification for the Area Chair (2/2)**

### **Summary of Reviewer Strengths and Weaknesses**

**Strengths:**

- Reviewer Rygs (score 8) highlights that *UniSVD achieves **strong performance*** compared to our primary baseline (i.e., the combined method) while remaining ***simple, training-free***, and ***easily plugs into various SVD-based methods without fine-tuning***.

- Reviewer B938 (score 4) acknowledges that our method is an *effective and practical approach for training-free model compression*, and recognizes the proposed technique as ***a novel and technically interesting idea***. The reviewer also notes that our extension to VLM tasks shows ***promising results*** and is ***indeed impressive***.

- Reviewer 3Es2 (score 4) highlights that our method ***successfully exploits the linear nature of attention mechanisms*** and reports ***substantial gains*** over multiple baseline methods.

- Reviewer 1opc (score 2) focuses on ***our discovery of asymmetric sensitivity between Q–K and V–O projections in attention***, evaluating this as ***a novel insight***.

**Weaknesses and Key Rebuttal Points:**

In this section, we summarize the key concerns raised by the reviewers and provide the corresponding rebuttal points.

**1. Verification of Generality (Reviewers 3Es2, 1opc, B938, Rygs)**

We demonstrated that UniSVD consistently yields effective compression across *a wide range of vision backbones, including DINOv2, ViT-Large, Swin-Large, and DeiT-Large*, in **Table 2** of the revised paper. To further validate generality, we applied our method to *segmentation, object detection, and vision–language tasks*, as reported in **Tables 4 and 5**, demonstrating strong generalizability beyond classification.

**2. Additional Overhead of Dynamic Selection (Reviewers 3Es2, 1opc, B938, Rygs)**

We clarified that the proposed dynamic selection does not incur any additional overhead during inference. We analyzed the latency at the pre-computation stage and provided detailed measurements. In response, *Reviewer B938 explicitly acknowledged that this concern was sufficiently addressed.*

**3. Generality for LLMs (Reviewers 1opc, B938, Rygs)**

In the rebuttal sections *“Discussion of UniSVD Limitations and Extensibility”* and *“Clarification on Applying UniSVD”*, we addressed the limitation that our method cannot be directly applied to current LLMs due to the absence of linear characteristics. In these sections, we also introduced emerging LLM designs with linear characteristics and discussed the potential extensibility of our method to them.

**4. Comparison with KO Ablation (Reviewer 1opc)**

We believe this concern was the primary reason for Reviewer 1opc's lower score. In our response for *"technical novelty related to fixed-selection ablation"*, we clarified that the KO ablation is an inherent part of our unilateral framework and already outperforms the combined method. As Reviewer 1opc also noted in the strengths, the KO ablation further demonstrates the asymmetric sensitivity, reinforcing our core insight.

**5. Additional Analyses Requested by Reviewers**

- **Robustness of dynamic selection across different ranks (Reviewer 3Es2):** We compared the dynamic selection method with fixed-selection cases across ranks, and added in **Figure 14 (left)** of the revised paper.

- **Selection ratio analysis (Reviewer 1opc):** We analyzed Q–K and V–O selection ratios and provided detailed results in **Table H** of the rebuttal and **Table 7** of the revised paper.

- **Application to additional SOTA methods (Reviewer 1opc):** We demonstrated that applying our method to the more recent study, SVD-LLM v2, achieves effective performance in  **Table Q**.

- **Evaluation on diverse hardware (Reviewer B938):** We included experiments across different GPU environments in **Table J** to show consistent efficiency.

- **Selection map visualization (Reviewer Rygs):** We added head-wise selection maps in **Figure 13** of the revised paper.

- **Performance-Rank curves (Reviewer Rygs):** We added detailed plots analyzing performance versus ranks in **Figure 14 (right)** of the revised paper.

- **Alternative error functions (Reviewer Rygs):** We experimented with additional error functions in **Table L**, and our results show that they can be effectively incorporated into our framework.

---

### **Summary of Discussion Phase**

Although we addressed all concerns raised by reviewers, only Reviewer B938 engaged further during discussion phase. Reviewer B938 emphasized the importance of evaluating the extensibility of our method to VLM tasks and requested additional benchmarks and latency analyses in VLM settings. In response, we provided the corresponding experimental results, and the reviewer also noted that our extension to VLM tasks shows ***promising results*** and is ***indeed impressive***. The reviewer also mentioned that the issue discussed in Common Response (3/3), “Additional Overhead of Dynamic Selection Compared to Fixed Selection,” was sufficiently addressed.

---

### Author Response · Authors · 2025-12-02
**Final Clarification for the Area Chair (1/2)**

Dear Area Chair,

We sincerely appreciate your efforts in guiding the review process despite the unforeseen circumstances surrounding the reassignment. We hope that the additional clarifications provided below will be helpful in supporting your evaluation of our submission.

---

### **Positioning Our Contribution in the Current Research Landscape**
Given the unexpected reassignment of submissions, we would like to provide a brief overview of the relevant research landscape and clarify the position of our work within it, to support the Area Chair’s assessment with full respect for AC’s expertise in this domain.

- *Combined Weight Decomposition Method* [1] (ICML 2023) :

This work was the first to propose a combined weight decomposition approach that explicitly leverages the linear operational characteristics of attention layers in vision models.

- *Per-weight Decomposition for LLMs* [2,3,4] (ICLR 2025), [5] (NAACL 2025) :

As weight decomposition has recently gained substantial interest in the LLM community, numerous methods have been proposed. These approaches target LLMs, whose attention projections cannot leverage linear characteristics, and therefore adopt per-weight decomposition strategies.

- *Per-weight Decomposition for Vision Models* [6] (CVPRW 2025) :

This work extends recent LLM-style weight decomposition methods to the vision domain, but still relies on a per-weight decomposition approach. In their comparison against the combined weight decomposition method [1], the authors report substantially lower performance at the same compression ratio because only the attention layers are reduced. Compared with prior work from three years ago [7], the results show a performance drop of about 20 %, indicating that the method fails to effectively leverage the linear characteristics of attention.

- *Unilateral Weight Decomposition Method (**Ours**)* :

We propose a novel and effective unilateral weight decomposition method for vision models that **leverages the linear characteristics of attention while improving upon the combined weight decomposition method**. In addition, we demonstrate that our approach can be used alongside existing per-weight decomposition techniques by applying ours to attention layers with linear operations, resulting in substantial performance improvements. Within this research context, we believe that our method and framework provide a practical foundation for guiding future directions in weight decomposition.

[1] Xiao, Jinqi, et al. "COMCAT: Towards Efficient Compression and Customization of Attention-Based Vision Models." International Conference on Machine Learning. PMLR, 2023.

[2] Wang, Xin, et al. "SVD-LLM: Truncation-aware Singular Value Decomposition for Large Language Model Compression." The Thirteenth International Conference on Learning Representations. 2025.

[3] Lin, Chi-Heng, et al. "MoDeGPT: Modular Decomposition for Large Language Model Compression." The Thirteenth International Conference on Learning Representations. 2025.

[4] Wang, Qinsi, et al. "Dobi-SVD: Differentiable SVD for LLM Compression and Some New Perspectives." The Thirteenth International Conference on Learning Representations. 2025.

[5] Wang, Xin, et al. "SVD-LLM V2: Optimizing Singular Value Truncation for Large Language Model Compression." Proceedings of the 2025 Conference of the Nations of the Americas Chapter of the Association for Computational Linguistics: Human Language Technologies (Volume 1: Long Papers). 2025.

[6] Thoma, Moritz, et al. "FLAR-SVD: Fast and Latency-Aware Singular Value Decomposition for Model Compression." Proceedings of the Computer Vision and Pattern Recognition Conference. 2025.

[7] Hsu, Yen-Chang, et al. "Language model compression with weighted low-rank factorization." International Conference on Learning Representations, 2022.

---

### **Summary of our Contribution**
We propose a novel attention weight decomposition method designed for vision models, where decomposition techniques can be particularly effective for attention layers involving linear operations. Building on this idea, our main contributions are as follows:

1. We propose a **novel and effective unilateral decomposition** approach that restructures the attention weights in a way that reduces computational overhead while preserving the essential representational capacity of the model.

2. We introduce a **dynamic selection method** that guarantees the minimum Frobenius norm error across different model architectures.

3. Finally, we show that this framework, when applied to attention layers **together with existing weight decomposition techniques**, yields substantial performance improvements.

---

### Meta-Review · Area_Chair_Sd6J · 2025-12-23

**Summary:**

Across reviews, the main concerns that informed my reject recommendation are:

UniSVD is viewed as a relatively small modification on top of standard SVD-based decomposition. Several reviewers questioned whether the dynamic selection adds meaningful novelty beyond a strong fixed choice.

The initial evaluation was largely confined to DeiT models on ImageNet classification, raising concerns about scalability to larger/recent backbones, other tasks, and multimodal settings.

Reviewers raised questions about end-to-end efficiency reporting, and whether the claimed speedups translate reliably across hardware and realistic settings.

Multiple reviewers asked for deeper analysis explaining why unilateral decomposition should preserve accuracy better than combined decomposition, and for more rigorous/quantitative support beyond qualitative singular-value observations.

**Reviewer Concerns:**

Concerns that were addressed by the rebuttal:

The rebuttal added experiments on larger/recent vision backbones, downstream tasks, and some VLM settings with additional benchmarks.

The authors clarified that dynamic selection is an offline precomputation step with no per-sample inference overhead, and provided wall-clock measurements for the selection pass.

Added latency on different GPU types and additional reporting such as VRAM/latency across batch size and sequence length, plus extra plots/ablations.

Explanation for parameter count differences and clarification of rank assignment strategy were added.

Concerns that are still outstanding:

Even after rebuttal, the method is still essentially a selection heuristic layered onto SVD decomposition. In several settings, a fixed selection performs nearly the same as the dynamic variant, which weakens the claimed methodological contribution.

Added analyses are largely empirical and do not provide a convincing or general explanation of when/why unilateral decomposition should dominate combined decomposition. The rebuttal itself acknowledges difficulty in fair singular-value comparisons across spaces and provides counter-examples where proxy errors do not predict accuracy.

The VLM latency results indicate that when the vision encoder is a small portion of total compute, end-to-end gains are small. Since the method is applied only to attention layers in the vision encoder, the overall practical impact can be limited in the most relevant large-scale settings.

While additional baselines/experiments were added, the evaluation still centers around SVD-style families. Broader comparisons are still limited, making it hard to conclude a clear state-of-the-art contribution.

**Reviewer Scores:**

Reviewer Rygs: 8–>8

Reviewer 3Es2: 4->4

Reviewer B938: 4->4

Reviewer 1opc: 2->3

---

### Decision · Program_Chairs · 2026-01-26

Reject